# Sparse Gaussian Processes: Structured Approximations and Power-EP Revisited

**Thang D. Bui**
Australian National University
`thang.bui@anu.edu.au`

**Michalis K. Titsias**
Google DeepMind
`mtitsias@google.com`

## Abstract

Inducing-point-based sparse variational Gaussian processes have become the standard workhorse for scaling up GP models. Recent advances show that these methods can be improved by introducing a diagonal scaling matrix to the conditional posterior density given the inducing points. This paper first considers an extension that employs a block-diagonal structure for the scaling matrix, provably tightening the variational lower bound. We then revisit the unifying framework of sparse GPs based on Power Expectation Propagation (PEP) and show that it can leverage and benefit from the new structured approximate posteriors. Through extensive regression experiments, we show that the proposed block-diagonal approximation consistently performs similarly to or better than existing diagonal approximations while maintaining comparable computational costs. Furthermore, the new PEP framework with structured posteriors provides competitive performance across various power hyperparameter settings, offering practitioners flexible alternatives to standard variational approaches.

## 1 Introduction

Gaussian processes (GPs) provide a principled framework for modelling functions that offer calibrated uncertainty and safeguard against overfitting, among many other benefits (see e.g., Rasmussen & Williams, 2006). However, their computational requirement, cubic in the number of training data $N$, is prohibitive for many practical applications. This bottleneck motivates the development of a plethora of scalable approximation methods (Quiñonero-Candela & Rasmussen, 2005; Liu et al., 2020), with sparse variational methods using inducing points arguably the most popular (Titsias, 2009; Hensman et al., 2013).

The key idea behind sparse variational GPs (SVGPs) is to approximate the posterior process using a small set of $M \ll N$ inducing points, reducing the computational complexity to $\mathcal{O}(NM^2)$ or $\mathcal{O}(M^3)$ in the batch and stochastic settings, respectively. A key assumption in the standard SVGP approximation is the prior distribution of the non-inducing function values conditioned on the inducing points remains unchanged in the approximate posterior, that is, $q(f_{\neq u}|u) = p(f_{\neq u}|u)$. Titsias (2025); Bui et al. (2025) recently showed that relaxing this assumption yields provably tighter variational bounds. In particular, the key innovation is slightly adjusting covariance of $q(f_{\neq u}|u)$ by a diagonal scaling matrix $\mathbf{M}$, leading to improved predictive performance while maintaining computational tractability. This approach has the original SVGP approach as a special case when $\mathbf{M} = \mathbf{I}$. Such improvement begs the question: can we achieve even better approximations by considering more expressive structures for $\mathbf{M}$ while preserving efficient computation?

To this end, we propose using block-diagonal structures for $\mathbf{M}$ and show that this choice leads to provably tighter variational bounds compared to existing diagonal approximations while maintaining the same computational complexity and ease of implementation. We then show that these structured approximations can also help with other inference schemes beyond variational inference. Specifically,

39th Conference on Neural Information Processing Systems (NeurIPS 2025).

certain structural choices for $\mathbf{M}$ lead to tractable Power Expectation Propagation (PEP) updates and approximate log marginal likelihood. This greatly extends and improves over the unifying framework of Bui et al. (2017).

The remainder of this paper is organised as follows. Section 2 reviews sparse variational GPs, recent advances in structured approximations, and the PEP framework for sparse GPs. Section 3 presents the proposed block-diagonal variational approximation. Section 4 extends the existing PEP framework with various structured posteriors. Section 5 evaluates the proposed methods on a suite of tasks. We then discuss related work in section 6 and conclude with a discussion of future directions in section 7.

## 2 Background

We first provide a summary of inducing-point sparse variational Gaussian processes (SVGP; Titsias, 2009; Hensman et al., 2013, 2015; Matthews et al., 2016), a recently proposed tighter bound (Bui et al., 2025; Titsias, 2025), and a power-EP based approach (Bui et al., 2017). Consider the supervised learning setting with an unknown input-output mapping $f$, a GP prior over this function $p(f|\gamma) = \mathcal{GP}(f; 0, k_\gamma)$, and a pointwise likelihood $p(\boldsymbol{y}|f, \mathbf{X}, \omega) = \prod_n p(y_n|f(\boldsymbol{x}_n), \omega)$, where $\mathbf{X} \in \mathbb{R}^{N \times D}$ and $\boldsymbol{y} \in \mathbb{R}^N$ are the training inputs and outputs, $k_\gamma$ is the covariance function governed by hyperparameters $\gamma$, and $\omega$ is the likelihood hyperparameters. In what follows, we will use $\theta$ to denote these hyperparameters and, when clear, drop the dependence on $\theta$ for brevity. Inference and learning in this model are computationally challenging for large-scale datasets due to the $\mathcal{O}(N^3)$ complexity; thus, efficient approximations are required. Sparse variational methods parameterise an approximate posterior based on $M$ inducing points, $\{\boldsymbol{z} \in \mathbb{R}^{M \times D}, \boldsymbol{u} \in \mathbb{R}^M\}$, with $M \ll N$, as follows,

$$q(f) = p(f_{\neq \boldsymbol{f}, \boldsymbol{u}}|\boldsymbol{f}, \boldsymbol{u})q(\boldsymbol{f}|\boldsymbol{u})q(\boldsymbol{u}), \tag{1}$$

where $\boldsymbol{f} = [f(\boldsymbol{x}_1), \cdots, f(\boldsymbol{x}_N)]$. Note that the factorisation here mirrors that in the prior, $p(f) = p(f_{\neq \boldsymbol{f}, \boldsymbol{u}}|\boldsymbol{f}, \boldsymbol{u})p(\boldsymbol{f}|\boldsymbol{u})p(\boldsymbol{u})$, where $p(\boldsymbol{u}) = \mathcal{N}(\boldsymbol{u}; \mathbf{0}, \mathbf{K_{uu}})$, $p(\boldsymbol{f}|\boldsymbol{u}) = \mathcal{N}(\boldsymbol{f}; \mathbf{K_{fu}}\mathbf{K_{uu}^{-1}}\boldsymbol{u}, \mathbf{D_{ff}})$, $\mathbf{D_{ff}} = \mathbf{K_{ff}} - \mathbf{Q_{ff}}$, $\mathbf{Q_{ff}} = \mathbf{K_{fu}}\mathbf{K_{uu}^{-1}}\mathbf{K_{uf}}$, $\mathbf{K_{ff}} = k(\mathbf{X}, \mathbf{X})$, $\mathbf{K_{fu}} = k(\mathbf{X}, \mathbf{z})$, and $\mathbf{K_{uu}} = k(\mathbf{z}, \mathbf{z})$. Note that we use $f$ to denote the function and $\boldsymbol{f}$ to denote the function values at the training inputs. The resulting variational lower bound to the log marginal likelihood is

$$\mathcal{F}_0 = -\mathrm{KL}[q(\boldsymbol{u})||p(\boldsymbol{u})] - \int q(\boldsymbol{u})\mathrm{KL}[q(\boldsymbol{f}|\boldsymbol{u})||p(\boldsymbol{f}|\boldsymbol{u})] + \sum_n \int q(\boldsymbol{u})q(f(\boldsymbol{x}_n)|\boldsymbol{u})\log p(y_n|f(\boldsymbol{x}_n)).$$

When $q(\boldsymbol{f}|\boldsymbol{u}) = p(\boldsymbol{f}|\boldsymbol{u}) = \mathcal{N}(\boldsymbol{f}; \mathbf{K_{fu}}\mathbf{K_{uu}^{-1}}\boldsymbol{u}, \mathbf{D_{ff}})$, the bound above becomes,

$$\mathcal{F}_1(q(\boldsymbol{u}), \theta) = -\mathrm{KL}[q(\boldsymbol{u})||p(\boldsymbol{u})] + \sum_n \int q(\boldsymbol{u})p(f(x_n)|\boldsymbol{u})\log p(y_n|f(x_n)), \tag{2}$$

commonly known as the uncollapsed SVGP bound (Hensman et al., 2015; Titsias, 2009). This bound conveniently allows both (i) tractable computation [$\mathcal{O}(NM^2)$ in the batch setting or $\mathcal{O}(BM^2 + M^3)$ where $B$ is the batch size in the mini-batch setting] and (ii) tractably handling of non-Gaussian likelihoods using quadrature or Monte Carlo estimation for the expected log-likelihood terms. For the Gaussian likelihood, the bound can be simplified to

$$\mathcal{F}_{1,r}(q(\boldsymbol{u}), \theta) = -\mathrm{KL}[q(\boldsymbol{u})||p(\boldsymbol{u})] + \sum_n \left[ \int q(\boldsymbol{u})\log \mathcal{N}(y_n; \mathbf{k}_{f_n\mathbf{u}}\mathbf{K_{uu}^{-1}}\boldsymbol{u}, \sigma^2) - \frac{d_{nn}}{2\sigma^2} \right], \tag{3}$$

where $\sigma^2$ is the observation noise and $d_{nn} = [\mathbf{D_{ff}}]_{nn}$. Furthermore, an optimal form for $q(\boldsymbol{u})$ can be found, $q(\boldsymbol{u}) \propto p(\boldsymbol{u})\mathcal{N}(\boldsymbol{y}; \mathbf{K_{fu}}\mathbf{K_{uu}^{-1}}\boldsymbol{u}, \sigma^2\mathbf{I}_N)$, yielding the following analytic collapsed bound (Titsias, 2009),

$$\mathcal{F}_{1,rc}(\theta) = \log \mathcal{N}(\boldsymbol{y}; \mathbf{0}, \mathbf{Q_{ff}} + \sigma^2\mathbf{I}_N) - \frac{1}{2\sigma^2}\sum_n d_{nn}. \tag{4}$$

The SVGP approach above has arguably been the most popular scalable GP approach in the literature. More recently, Bui et al. (2025); Titsias (2025) show that this approach can be improved by relaxing the $q(\boldsymbol{f}|\boldsymbol{u}) = p(\boldsymbol{f}|\boldsymbol{u})$ assumption. Specifically, when $q(\boldsymbol{f}|\boldsymbol{u}) = \mathcal{N}(\boldsymbol{f}; \mathbf{K_{fu}}\mathbf{K_{uu}^{-1}}\boldsymbol{u}, \mathbf{D_{ff}^{1/2}}\mathbf{M}\mathbf{D_{ff}^{1/2}})$,

where $\mathbf{M} = \mathrm{diag}([m_1, \ldots, m_N])$, the uncollapsed and collapsed bounds in the regression case are:

$$\mathcal{F}_{2,r}(q(\boldsymbol{u}), \theta) = -\mathrm{KL}[q(\boldsymbol{u})\|p(\boldsymbol{u})] + \sum_n \left[ \int q(\boldsymbol{u}) \log \mathcal{N}(y_n; \mathbf{k}_{f_n\mathbf{u}}\mathbf{K}_{\mathbf{uu}}^{-1}\boldsymbol{u}, \sigma^2) - \frac{1}{2}\log\left(1 + \frac{d_{nn}}{\sigma^2}\right) \right] \quad (5)$$

$$\mathcal{F}_{2,rc}(\theta) = \log \mathcal{N}(\boldsymbol{y}; \mathbf{0}, \mathbf{Q}_{\mathbf{ff}} + \sigma^2 \mathbf{I}_N) - \frac{1}{2}\sum_n \log\left(1 + \frac{d_{nn}}{\sigma^2}\right). \quad (6)$$

Note that the optimal form for $m_n$ is $m_n = \sigma^2/(\sigma^2 + d_{nn}) < 1$; and eqs. (5) and (6) are tighter than eqs. (3) and (4) for fixed $\theta$ and $q(\boldsymbol{u})$ since $\log\left(1 + d_{nn}/\sigma^2\right) \leq d_{nn}/\sigma^2$.

The posterior approximation in eq. (1) can also be used in other deterministic inference strategies. For example, in the regression case, for $q(\boldsymbol{f}|\boldsymbol{u}) = p(\boldsymbol{f}|\boldsymbol{u})$, Bui et al. (2017) showed that Power-Expectation Propagation (PEP) yields an analytic collapsed approximate marginal likelihood,

$$\mathcal{F}_{3,rc}(\theta) = \log \mathcal{N}(\boldsymbol{y}; \mathbf{0}, \mathbf{Q}_{\mathbf{ff}} + \alpha\mathbf{D}_{\mathbf{ff}} + \sigma^2 \mathbf{I}_N) - \frac{1-\alpha}{2\alpha}\sum_n \log\left(1 + \alpha\frac{d_{nn}}{\sigma^2}\right), \quad (7)$$

and a closed form $q(\boldsymbol{u})$, $q(\boldsymbol{u}) \propto p(\boldsymbol{u})\mathcal{N}(\boldsymbol{y}; \mathbf{K}_{\mathbf{fu}}\mathbf{K}_{\mathbf{uu}}^{-1}\boldsymbol{u}, \alpha\mathbf{D}_{\mathbf{ff}} + \sigma^2 \mathbf{I}_N)$, where $\alpha$ is the power hyperparameter in PEP. This framework encompasses a multitude of approximations, such as the SVGP approximation (as $\alpha \to 0$) and FITC (Snelson & Ghahramani, 2005; Qi et al., 2010) ($\alpha = 1$).

## 3    A block-diagonal structured variational approximation

We first consider the following posterior approximation:

$$q(f) = p(f_{\neq \boldsymbol{f}, \boldsymbol{u}}|\boldsymbol{f}, \boldsymbol{u})q(\boldsymbol{f}|\boldsymbol{u})q(\boldsymbol{u}), \quad q(\boldsymbol{f}|\boldsymbol{u}) = \mathcal{N}(\boldsymbol{f}; \mathbf{K}_{\mathbf{fu}}\mathbf{K}_{\mathbf{uu}}^{-1}\boldsymbol{u}, \mathbf{C}),$$

where we have not posited a form for the covariance $\mathbf{C}$. Interestingly, this leads to the familiar optimal form for $q(\boldsymbol{u})$, $q(\boldsymbol{u}) \propto p(\boldsymbol{u})\mathcal{N}(\boldsymbol{y}; \mathbf{K}_{\mathbf{fu}}\mathbf{K}_{\mathbf{uu}}^{-1}\boldsymbol{u}, \sigma^2 \mathbf{I}_N)$. The resulting collapsed bound is,

$$\mathcal{F}(\theta) = \log \mathcal{N}(\boldsymbol{y}; 0, \mathbf{Q}_{\mathbf{ff}} + \sigma^2 \mathbf{I}_N) - \frac{1}{2}\mathrm{trace}[(\sigma^{-2}\mathbf{I}_N + \mathbf{D}_{\mathbf{ff}}^{-1})\mathbf{C}] - \frac{1}{2}\log|\mathbf{C}^{-1}\mathbf{D}_{\mathbf{ff}}| + \frac{N}{2}.$$

Except for some special cases, the bound above is as expensive as the original log marginal likelihood to compute. Specifically, as shown in the background, $\mathbf{C} = \mathbf{D}_{\mathbf{ff}}^{1/2}\mathbf{M}\mathbf{D}_{\mathbf{ff}}^{1/2}$ with $\mathbf{M} = \mathbf{I}_N$ (Titsias, 2009) or $\mathbf{M} = m\mathbf{I}_N$ (Artemev et al., 2021) or $\mathbf{M} = \mathrm{diag}(\{m_n\}_{n=1}^N)$ (Titsias, 2025; Bui et al., 2025) admit tractability, and each move (from $\mathbf{I}_N$ to $m\mathbf{I}_N$, and from $m\mathbf{I}_N$ to $\mathrm{diag}(\{m_n\}_{n=1}^N)$) makes the bound tighter. It is thus natural to enquire what structure to encode in $\mathbf{M}$ to further improve the bound, retain tractable computation, and potentially improve predictive performance.

We now consider one such structure, a block-diagonal $\mathbf{M}$, $\mathbf{M} = \mathrm{blkdiag}(\{\mathbf{m}_b\}_{b=1}^B)$, where $B$ is the number of blocks and $\mathbf{m}_b \in \mathbb{R}^{N_b \times N_b}$. Substituting this into the bound above gives

$$\mathcal{F}_4(\theta) = \log \mathcal{N}(\boldsymbol{y}; 0, \mathbf{Q}_{\mathbf{ff}} + \sigma^2 \mathbf{I}_N) - \frac{1}{2}\sum_b \left[ \frac{1}{\sigma^2}\mathrm{trace}[\mathbf{m}_b\mathbf{D}_{\mathbf{f}_b\mathbf{f}_b}] + \mathrm{trace}[\mathbf{m}_b] - \log|\mathbf{m}_b| - N_b \right].$$

We can obtain the optimal $\mathbf{m}_b$, $\mathbf{m}_b = (\mathbf{I}_b + \sigma^{-2}\mathbf{D}_{\mathbf{f}_b\mathbf{f}_b})^{-1}$, leading to the following collapsed bound,

$$\mathcal{F}_{4,rc}(\theta) = \log \mathcal{N}(\boldsymbol{y}; 0, \mathbf{Q}_{\mathbf{ff}} + \sigma^2 \mathbf{I}_N) - \frac{1}{2}\sum_b \log|\mathbf{I}_b + \sigma^{-2}\mathbf{D}_{\mathbf{f}_b\mathbf{f}_b}|. \quad (8)$$

Due to the Hadamard's inequality, $|\mathbf{I}_b + \sigma^{-2}\mathbf{D}_{\mathbf{f}_b\mathbf{f}_b}| < \prod_i(1 + \sigma^{-2}[\mathbf{D}_{\mathbf{f}_b\mathbf{f}_b}]_{ii})$, and thus $\log|\mathbf{I}_b + \sigma^{-2}\mathbf{D}_{\mathbf{f}_b\mathbf{f}_b}| < \sum_i \log(1 + \sigma^{-2}[\mathbf{D}_{\mathbf{f}_b\mathbf{f}_b}]_{ii})$. In other words, the bound in eq. (8) [$\mathbf{M}$ is block-diagonal] is provably tighter than the bound in eq. (6) [$\mathbf{M}$ is diagonal].

Similar to the standard SVGP approach, for large datasets, it is more convenient to work with the following uncollapsed bound that supports stochastic optimisation,

$$\mathcal{F}_{4,r}(\cdot) = -\mathrm{KL}[q(\boldsymbol{u})\|p(\boldsymbol{u})] + \sum_{b=1}^B \left[ \int q(\boldsymbol{u}) \log \mathcal{N}(\boldsymbol{y}_b; \mathbf{K}_{\mathbf{f}_b\mathbf{u}}\mathbf{K}_{\mathbf{uu}}^{-1}\boldsymbol{u}, \sigma^2 \mathbf{I}_b) - \frac{1}{2}\log|\mathbf{I}_b + \sigma^{-2}\mathbf{D}_{\mathbf{f}_b\mathbf{f}_b}| \right]. \quad (9)$$

If the B blocks are of roughly equal size, computing the bound in eq. (9) using the entire training set takes $\mathcal{O}(M^3 + NM^2 + B[N/B]^3)$. However, in practice, we perform stochastic optimisation,

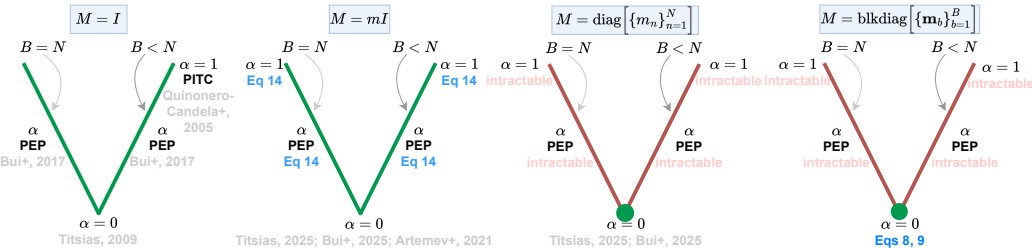

Figure 1: Connections between the sparse GP regression methods from the Power-EP perspective. Green means computationally tractable, red means intractable, and blue represents the new methods presented in this paper. $B < N$ means the training points are partitioned into B disjoint blocks. $B = N$ means having the same number of blocks as training points, i.e., block size equal to 1.

where we unbiasedly approximate the sum over blocks in eq. (9) using one random block to obtain the stochastic bound

$$\widetilde{\mathcal{F}}_{4,r}(\cdot) = -\text{KL}[q(\boldsymbol{u})||p(\boldsymbol{u})] + B\left[\int q(\boldsymbol{u})\log\mathcal{N}(\boldsymbol{y}_b; \mathbf{K}_{\mathbf{f}_b\mathbf{u}}\mathbf{K}_{\mathbf{uu}}^{-1}\boldsymbol{u}, \sigma^2\mathbf{I}_b) - \frac{1}{2}\log|\mathbf{I}_b + \sigma^{-2}\mathbf{D}_{\mathbf{f}_b\mathbf{f}_b}|\right], \quad (10)$$

based on which we perform stochastic gradient updates by cycling over the $B$ blocks. If we judiciously choose the block size $\frac{N}{B}$ to be $M$ (i.e., block size equals to the number of inducing points), the computational requirement per iteration is only $\mathcal{O}(M^3)$. Therefore, eq. (10) has a small implementation overhead compared to standard stochastic sparse GP objectives. The precise extra overhead involves taking the Cholesky decomposition of $\mathbf{I}_b + \sigma^{-2}\mathbf{D}_{\mathbf{f}_b\mathbf{f}_b}$, needed when computing the log-determinant regularisation term.

We will next consider a special case. When we let all $\mathbf{m}_b$ matrices to be the same, $\mathbf{m}_b = \mathbf{m}$, we arrive at the optimal $\mathbf{m}$, $\mathbf{m} = (\mathbf{I}_b + B^{-1}\sigma^{-2}\sum_b \mathbf{D}_{\mathbf{f}_b\mathbf{f}_b})^{-1}$, and the resulting collapsed bound,

$$\mathcal{F}_5(\theta) = \log\mathcal{N}(\boldsymbol{y}; 0, \mathbf{Q}_{\mathbf{ff}} + \sigma^2\mathbf{I}_N) - \frac{B}{2}\log|\mathbf{I}_b + \frac{1}{B\sigma^2}\sum_b \mathbf{D}_{\mathbf{f}_b\mathbf{f}_b}|. \quad (11)$$

Since the log-determinant is a concave function on the cone of positive definite matrices, we can apply Jensen's inequality to show that the bound above is less tight compared to eq. (8). As the block size equals one, this becomes the *spherical* bound in Titsias (2025); Artemev et al. (2021).

A disadvantage of diagonal and block diagonal structures in $\mathbf{M}$ is the expensive predictive covariance. However, we can approximate it by reverting to using $q(\boldsymbol{f}|\boldsymbol{u}) \approx p(\boldsymbol{f}|\boldsymbol{u})$ at test time. Bui et al. (2025) noted that this approximation does not degrade the performance compared to the expensive exact predictive distribution. In other words, in practice, we only use the new structured posterior in training, and therefore, any improvement in predictive performance at test time will come from better $q(\boldsymbol{u})$ and hyperparameters.

## 4 A more general approximation based on Power Expectation Propagation

Although the variational sparse GP approach has captured the spotlight in the sparse GP literature, Bui et al. (2017) showed various variants of PEP can be as competitive or better. We will now revisit the framework of Bui et al. (2017) and explore how it can be improved by leveraging the recent innovation in structured posterior approximations (Titsias (2025); Bui et al. (2025) and section 3) originally developed in the variational inference setting. We first write down the joint density of the exact model and the approximate posterior,

$$p(f, \boldsymbol{y}) = p(f_{\neq\boldsymbol{f},\boldsymbol{u}}|\boldsymbol{f},\boldsymbol{u})p(\boldsymbol{f}|\boldsymbol{u})p(\boldsymbol{u})\prod_{n=1}^{N} p(y_n|\boldsymbol{f}_n) \quad (12)$$

$$q(f) \propto p(f_{\neq\boldsymbol{f},\boldsymbol{u}}|\boldsymbol{f},\boldsymbol{u})q(\boldsymbol{f}|\boldsymbol{u})p(\boldsymbol{u})\prod_{b=1}^{B} t_b(\boldsymbol{u}) \quad (13)$$

where the $N$ training points are partitioned into $B$ disjoint blocks, and the factors $t_b(\boldsymbol{u})$ are assumed to be Gaussian. Instead of using $q(\boldsymbol{f}|\boldsymbol{u}) = p(\boldsymbol{f}|\boldsymbol{u})$ as in Bui et al. (2017), we consider $q(\boldsymbol{f}|\boldsymbol{u}) = \mathcal{N}(\boldsymbol{f}; \mathbf{K_{fu}K_{uu}^{-1}}\boldsymbol{u}; \mathbf{D_{ff}^{1/2}MD_{ff}^{1/2}})$ where $\mathbf{M} = \mathrm{blkdiag}(\{\mathbf{m}_b\}_{b=1}^B)$, that is, the blocks in $\mathbf{M}$ match that of the likelihood partitions.

The PEP procedure (Minka, 2004) iteratively updates $t_b(\boldsymbol{u})$ by (i) first remove an $\alpha$-fraction of $t_b(\boldsymbol{u})$ from $q(f)$ to form the cavity distribution, $q^{\backslash b}(f) = q(f)/t_b^\alpha(\boldsymbol{u})$, (ii) incorporate an $\alpha$-fraction of the likelihood for the $b$-th block $p(\boldsymbol{y}_b|\boldsymbol{f}_b) = \prod_{n=1}^{N_b} p(y_n|f_n)$ to form the tilted distribution, $\tilde{q}(f) = q^{\backslash b}(f)p^\alpha(\boldsymbol{y}_b|\boldsymbol{f}_b)$, (iii) find a new approximation $q(f)$ that minimises $\mathrm{KL}[\tilde{q}(f)||q(f)]$, and (iv) adjust $t_b(\boldsymbol{u})$ based on the new posterior using $t_b(\boldsymbol{u}) = [q(f)/q^{\backslash b}(f)]^{1/\alpha}$ or $t_b(\boldsymbol{u}) \leftarrow t_b^{1-\alpha}(\boldsymbol{u})[q(f)/q^b(f)]$. These steps are repeated for all blocks until convergence. Readers might have noticed that step (iii) is a daunting task as it involves moment matching for the entire Gaussian processes; however, due to the structure of the approximate posterior $q(f)$, it is sufficient to perform moment matching for the finite function values $\boldsymbol{u}$ (Bui et al., 2017). In addition, this procedure returns an estimate of the log marginal likelihood that can be used for hyperparameter optimisation.

Mirroring the derivation in Bui et al. (2017), we can show the optimal form for $t_b(\boldsymbol{u})$ has rank $N_b = |\boldsymbol{f}_b|$, $t_b(\boldsymbol{u}) = \mathcal{N}(\mathbf{K_{f_bu}K_{uu}^{-1}}\boldsymbol{u}; \boldsymbol{g}_b, \boldsymbol{v}_b)$. In the regression case, $\boldsymbol{g}_b = \boldsymbol{y}_b$ and $\boldsymbol{v}_b = \alpha[\mathbf{D_{ff}^{1/2}MD_{ff}^{1/2}}]_{bb} + \sigma^2\mathbf{I}_b$. The full derivation is rather lengthy and is included in the appendix; however, one can verify that this is a stable fixed point of the procedure by noting that the $\alpha$-fraction of $t_b(\boldsymbol{u})$ is identical to the contribution of $p^\alpha(y_b|\boldsymbol{f}_b)$ to the posterior at $\boldsymbol{u}$, $\int d\boldsymbol{f}_b q(\boldsymbol{f}_b|\boldsymbol{u})p^\alpha(\boldsymbol{y}_b|\boldsymbol{f}_b)$. The optimal $q(\boldsymbol{u})$ is thus $q(\boldsymbol{u}) \propto p(\boldsymbol{u})\mathcal{N}(\boldsymbol{y}; \mathbf{K_{fu}K_{uu}^{-1}}\boldsymbol{u}, \alpha\mathrm{blkdiag}(\{[\mathbf{D_{ff}^{1/2}MD_{ff}^{1/2}}]_{bb}\}_{b=1}^B) + \sigma^2\mathbf{I}_N)$. Furthermore, for the regression case, we can further derive the approximate marginal likelihood,

$$\mathcal{F}_{5,rc}(\theta, \mathbf{M}) = \log\mathcal{N}(\boldsymbol{y}; \mathbf{0}, \mathbf{Q_{ff}} + \alpha\mathrm{blkdiag}(\{[\mathbf{D_{ff}^{1/2}MD_{ff}^{1/2}}]_{bb}\}_{b=1}^B) + \sigma^2\mathbf{I}_N)$$
$$+ \sum_b \left[ -\frac{1-\alpha}{2\alpha}\log\left|\mathbf{I}_b + \alpha\frac{[\mathbf{D_{ff}^{1/2}MD_{ff}^{1/2}}]_{bb}}{\sigma^2}\right| - \frac{1}{2\alpha}\log|\mathbf{I}_b + \alpha(\mathbf{m}_b - \mathbf{I}_b)| + \frac{1}{2}\log|\mathbf{m}_b| \right].$$

We note that, for a general $\alpha$ and $\mathbf{M}$, including diagonal and block-diagonal cases, the PEP procedure above as well the approximate marginal likelihood for the regression case is computationally intractable (i.e., cubic in $N$) due to the need to find the (block-)diagonal of $\mathbf{D_{ff}^{1/2}MD_{ff}^{1/2}}$. We will now discuss the tractable special cases.

**Remark 1** *When $\mathbf{M}$ is diagonal or block-diagonal, the approximate marginal likelihood and posterior approximation above are only tractable as $\alpha \to 0$. Specifically, when $\mathbf{M} = \mathrm{diag}(\{m_n\}_{n=1}^N)$, the objective becomes the variational bound of Titsias (2025); Bui et al. (2025), and when $\mathbf{M} = \mathrm{blkdiag}(\{\mathbf{m}_b\}_{b=1}^B)$, the objective matches the variational bound in eq. (8).*

**Remark 2** *When $\mathbf{M} = m\mathbf{I}_N$, the approximate marginal likelihood and posterior approximation are computationally tractable for all $\alpha$'s. In particular, the optimal $q(\boldsymbol{u})$ is $q(\boldsymbol{u}) \propto p(\boldsymbol{u})\mathcal{N}(\boldsymbol{y}; \mathbf{K_{fu}K_{uu}^{-1}}\boldsymbol{u}, m\alpha\mathrm{blkdiag}(\{\mathbf{D_{f_bf_b}}\}_{b=1}^B) + \sigma^2\mathbf{I}_N)$, and the approximate marginal likelihood becomes,*

$$\mathcal{F}_6(\theta, m) = \log\mathcal{N}(\boldsymbol{y}; 0, \mathbf{Q_{ff}} + m\alpha\mathrm{blkdiag}(\{\mathbf{D_{f_bf_b}}\}_{b=1}^B) + \sigma^2\mathbf{I}_N)$$
$$- \frac{1-\alpha}{2\alpha}\sum_b \left[ \log\left|\mathbf{I}_b + \frac{\alpha m\mathbf{D_{f_bf_b}}}{\sigma^2}\right| \right] - \frac{N}{2\alpha}\log(1 + \alpha(m-1)) + \frac{N}{2}\log(m). \quad (14)$$

In this special case, we note the following. First, as a sanity check, we can see that when $m = 1$, we recover the Power-EP approximate marginal likelihood of Bui et al. (2017):

$$\mathcal{F}_{6,m=1}(\theta) = \log\mathcal{N}(\boldsymbol{y}; 0, \mathbf{Q_{ff}} + \alpha\mathrm{blkdiag}(\mathbf{D_{ff}}) + \sigma^2\mathbf{I}_N) - \frac{1-\alpha}{2\alpha}\sum_n \log\left|\mathbf{I}_b + \alpha\frac{\mathbf{D_{f_bf_b}}}{\sigma^2}\right|.$$

Second, when $\alpha = 1$, the objective in eq. (14) becomes $\mathcal{F}_{6,\alpha=1}(\theta, m) = \log\mathcal{N}(\boldsymbol{y}; 0, \mathbf{Q_{ff}} + m\mathrm{blkdiag}(\mathbf{D_{ff}}) + \sigma^2\mathbf{I}_N)$. When $m = 1$, this becomes the PITC marginal likelihood and for block size equal to one it further reduces to FITC (Quiñonero-Candela & Rasmussen, 2005).

Third, as $\alpha \to 0$, we recover the *spherical* bound in (Bui et al., 2025; Titsias, 2025; Artemev et al., 2021). Only in this setting, we can derive the optimal $m = (1 + N^{-1}\sum_n d_n/\sigma^2)^{-1}$.

Finally, inspired by the uncollapsed variational bound, we can optimise an uncollapsed version of eq. (14) that supports stochastic optimisation as follows,

$$\mathcal{F}_{6,r}(q(\boldsymbol{u}), \theta, \mathbf{M}) = -\mathrm{KL}[q(\boldsymbol{u})||p(\boldsymbol{u})] + \sum_b \left[ \int q(\boldsymbol{u}) \log \mathcal{N}(\boldsymbol{y}_b; \mathbf{K}_{\mathbf{f}_b \mathbf{u}} \mathbf{K}_{\mathbf{uu}}^{-1} \boldsymbol{u}, m\alpha \mathbf{D}_{\mathbf{f}_b \mathbf{f}_b} + \sigma^2 \mathbf{I}_b) \right]$$

$$- \frac{1-\alpha}{2\alpha} \sum_b \left[ \log \left| \mathbf{I}_b + \frac{\alpha m \mathbf{D}_{\mathbf{f}_b \mathbf{f}_b}}{\sigma^2} \right| \right] - \frac{N}{2\alpha} \log (1 + \alpha(m-1)) + \frac{N}{2} \log(m).$$

That is, instead of running the PEP procedure, we can optimise the objective above to yield the same fixed point as PEP. We attempt to visualise the connections between the methods, the special cases and the broader literature in fig. 1.

## 5 Experiments

Having described the new block-diagonal structure in sparse variational GPs and revisited the unified work of Bui et al. (2017) in light of the new approximate posteriors, we will detail the experiments to qualitatively investigate (i) if the proposed block-diagonal approximation in section 3 yields better performance and, if yes, how, and (ii) whether having $m \neq 1$ benefits power expectation propagation in section 4 the same way it does to variational inference. All experiments were done on either a V100 GPU or a MacBook laptop. We provide an implementation here `https://github.com/thangbui/tighter_sparse_gp`.

### 5.1  1-D regression and biases in hyperparameter estimation

We first illustrate the difference between the proposed and existing methods on a simple 1D regression problem (Snelson & Ghahramani, 2005). In particular, we compare Titsias' collapsed bound in eq. (4) [SGPR], the bound of Titsias (2025); Bui et al. (2025) in eq. (6) [T-SGPR], the bound with block diagonal $\mathbf{M}$ in eq. (8) with 10 and 20 blocks [20 and 10 data points per block, respectively, BT-SGPR], the PEP approach of Bui et al. (2017) with $\alpha = 0.5$ [PEP], and the PEP approach in eq. (14) with $B = N$ and $\alpha = 0.5$ [T-PEP]. We used 5 inducing points in this experiment. The key results are summarised in fig. 2. It can be observed that (i) the block-diagonal approximation improves over the diagonal one in this example, (ii) increasing the number of training points in each block tightens the bound, (iii) the structured posterior approximation also helps in PEP, and (iv) hyperparameter optimisation using a more structured approximation tend to result in a smaller noise variance and a larger kernel variance. We note that the PEP approximate marginal likelihood is not guaranteed to be a lower bound and therefore optimising it can result in pathological behaviours, for example, when $\alpha = 1$, the noise variance can be severely underestimated (Bauer et al., 2016).

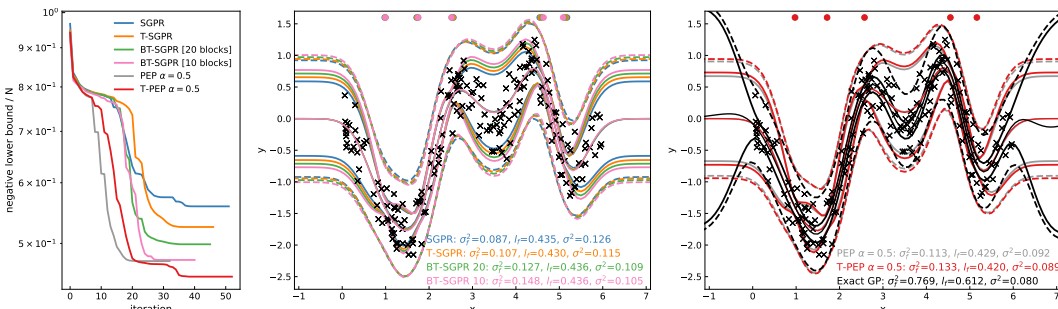

Figure 2: *Left:* Variational bounds during training on the Snelson dataset. *Middle and right:* Predictive mean and intervals using various methods and the final hyperparameter values.

To further investigate point (iv), we picked a subset of the KIN40K dataset with 5,000 data points, and ran an experiment to compare the sparse approximations with exact GP. For each method, we recorded in table 1 the exact or approximate log marginal likelihood, the predictive performance measured by root mean squared error (RMSE) and log likelihood (LL), the noise standard deviation $\sigma$ and the kernel lengthscales. Similar to the observation in the Snelson dataset above, the noise estimate is smaller when moving from $\mathbf{M} = \mathbf{I}_N$ to increasingly more structured $\mathbf{M}$, translating

to better predictions. The trend seems to be consistent across two numbers of inducing points. In addition, there is no notable difference in the lengthscales between PEP, T-PEP, and the structured variational approximations; however, these methods tend to *leverage* more dimensions than SGPR for $M = 256$.

Table 1: Exact/approximate marginal likelihoods, predictive performance, and lengthscales given by various methods on 5,000 samples from the KIN40K dataset.

| Method | M = 256 | | | | | M = 512 | | | | |
|--------|------|------|------|----------|-------------|------|------|------|----------|-------------|
| | Obj. | RMSE | LL | $\sigma$ | lengthscales | Obj. | RMSE | LL | $\sigma$ | lengthscales |
| Exact | -0.66 | 0.12 | 0.80 | 0.00 | | -0.66 | 0.12 | 0.80 | 0.00 | |
| SGPR | 0.88 | 0.26 | -0.14 | 0.30 | | 0.66 | 0.22 | 0.02 | 0.25 | |
| T-SGPR | 0.78 | 0.22 | -0.06 | 0.26 | | 0.51 | 0.18 | 0.11 | 0.21 | |
| BT-SGPR [50] | 0.75 | 0.22 | -0.05 | 0.25 | | 0.50 | 0.18 | 0.12 | 0.20 | |
| BT-SGPR [10] | 0.66 | 0.20 | -0.03 | 0.23 | | 0.44 | 0.17 | 0.13 | 0.19 | |
| PEP [0.5] | 0.66 | 0.23 | -0.02 | 0.22 | | 0.42 | 0.20 | 0.14 | 0.19 | |
| T-PEP [0.5] | 0.48 | 0.20 | 0.02 | 0.18 | | 0.18 | 0.16 | 0.19 | 0.14 | |

## 5.2 Block-diagonal structured variational approximation

We next ran an experiment to validate the utility of the proposed block-structured approximation in section 3 on four real-world regression datasets[1]. For each dataset and each inducing point configuration ($M = 256$ or $M = 512$), we compare the uncollapsed variational bounds of Titsias (2009); Hensman et al. (2015) [eq. (3), SVGP], Titsias (2025); Bui et al. (2025) [eq. (5), T-SVGP], and the proposed bound in eq. (9) [BT-SVGP], corresponding to $\mathbf{M} = \mathbf{I}_N$, $\mathbf{M} = \mathrm{diag}(\{m_n\}_{n=1}^N)$, and $\mathbf{M} = \mathrm{blkdiag}(\{\mathbf{m}_b\}_{b=1}^B)$, respectively. We repeated the experiment 10 times, each using a random train/test split, a batch size of 500 (also the block size), random partitioning of the training data into blocks, and 300 epochs for training. The average variational bound (ELBO) and test performance after training are shown in fig. 3. Similar to the earlier experiments, the benefit of the block-structured approximation is also clearly demonstrated here: it tightens the variational bound compared to that of the diagonal $\mathbf{M}$ and consistently yields comparable or better predictive performance. We note again that (i) the estimated observation noise tends to be smaller when employing the new bound (see the appendix), and (ii) there is a minimal implementation overhead compared to Titsias (2009, 2025); Bui et al. (2025) to result in these gains.

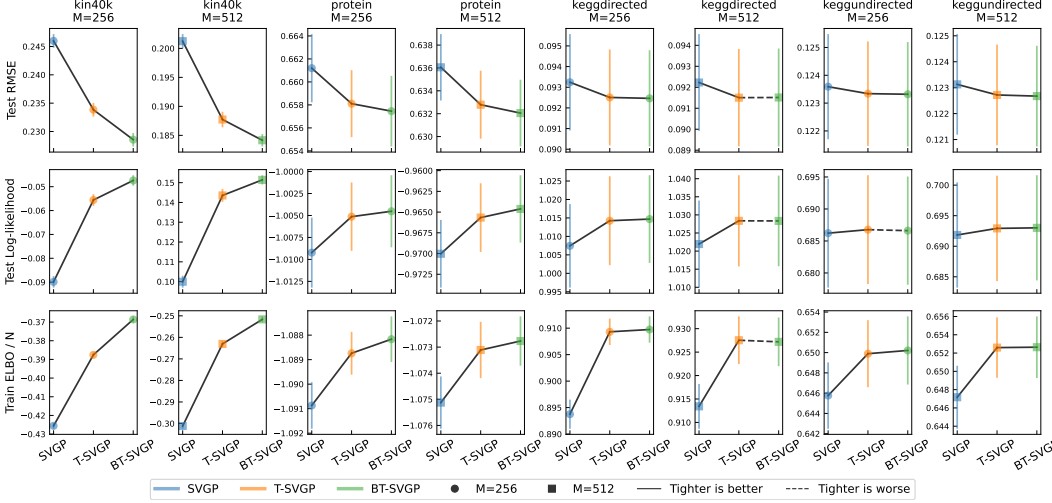

Figure 3: Lower bounds (ELBO) and predictive performance of various variational methods with $\mathbf{M} = \mathbf{I}_N$ [SVGP], $\mathbf{M} = \mathrm{diag}(\{m_n\}_{n=1}^N)$ [T-SVGP], and $\mathbf{M} = \mathrm{blkdiag}(\{\mathbf{m}_b\}_{b=1}^B)$ [BT-SVGP].

---

[1]We used the splits available in this repository `https://github.com/treforevans/uci_datasets`.

## 5.3 Power-EP with a structured approximate posterior [$\mathbf{M} = m\mathbf{I}_N$]

As shown in section 4, the structured approximate posterior considered by Titsias (2025) can be utilised in PEP and in the regression case, the approximate posterior and marginal likelihood are analytically available. To evaluate its practical utility, we ran an experiment on five small regression datasets, comparing the PEP approach of Bui et al. (2017) [$\mathbf{M} = \mathbf{I}_N$] to the proposed approach in section 4 [$\mathbf{M} = m\mathbf{I}_N$]. The typical performance across various inducing point configurations is shown in fig. 4, with the full results included in the appendix. It is noticeable that the Power-EP scheme with $m \neq 1$ tends to outperform the corresponding setting when $m = 1$. To elucidate the trend, we plot the difference between the performance of $\mathbf{M} = \mathbf{I}_N$ and $\mathbf{M} = m\mathbf{I}_N$ in fig. 5. We note that $m \neq 1$ outperforms $m = 1$ on all datasets in terms of RMSE, but log-likelihood performance degrades when $\alpha$ is closer to 1. These results suggest that for $m \neq 1$, intermediate $\alpha$ values such as 0.5 are most competitive in terms of both RMSE and LL, in line with recommendations from Bui et al. (2017) when $m = 1$.

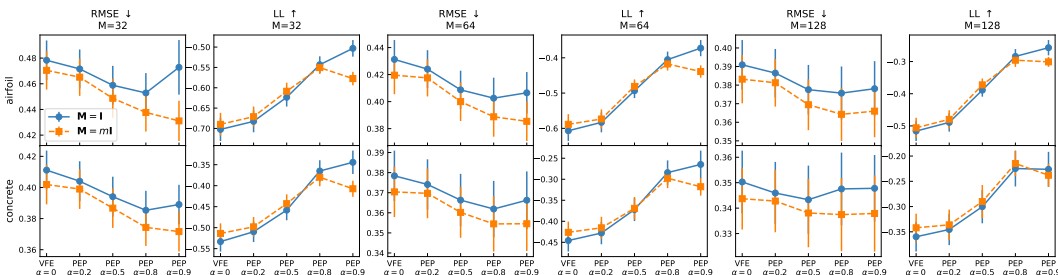

Figure 4: Predictive performance of power expectation propagation with $\mathbf{M} = \mathbf{I}_N$ and $\mathbf{M} = m\mathbf{I}_N$ on two UCI datasets. Results for other datasets are in the appendix.

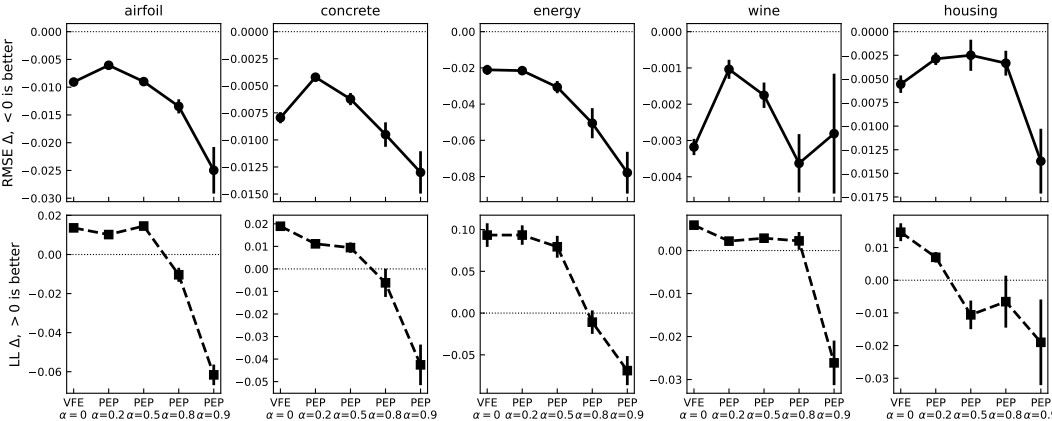

Figure 5: Difference in PEP performance between $\mathbf{M} = \mathbf{I}_N$ and $\mathbf{M} = m\mathbf{I}_N$ on five UCI datasets.

## 6 Related work

The use of inducing points for sparse approximations in Gaussian processes has a rich history, to name a few approaches, sparse online GPs (Csató & Opper, 2002), DTC (Seeger et al., 2003), FITC approximation (Snelson & Ghahramani, 2005), and PITC (Quiñonero-Candela & Rasmussen, 2005). The most notable was the variational approach of Titsias (2009), who introduced a principled method for selecting inducing points by optimising a variational lower bound. Hensman et al. (2013, 2015) extended this approach to enable stochastic optimisation and non-Gaussian likelihoods, significantly broadening the applicability of sparse GPs to large datasets. Other work on inducing point methods have exploited Kronecker products (Wilson & Nickisch, 2015), nearest neighbour structures (Tran et al., 2021; Wu et al., 2022) and inter-domain inducing points (Lázaro-Gredilla &

Figueiras-Vidal, 2009; Hensman et al., 2018). Also, recent theoretical work (Burt et al., 2020) studied the approximation convergence with respect to the number of inducing points.

Our work is most closely related to the recent advances by Titsias (2025); Bui et al. (2025), who showed that relaxing the standard assumption with diagonal scaling matrices improves the variational bound. Our block-diagonal extension naturally builds upon this line of work, showing practical benefits. Similarly, our extension of the PEP framework builds directly on Bui et al. (2017), expanding their unifying perspective by incorporating structured posterior approximations. Note that our work is distinct from the PITC approximation. PITC is derived from the prior modification perspective, where the prior is modified so that the blocks of function values are conditionally independent given the inducing points. Bui et al. (2017) showed that this is equivalent to EP when retaining the prior conditional in the approximate posterior, which differs from our proposed structured conditional distribution.

A key component in the sparse GP approximate posterior is $q(\boldsymbol{u})$, and imposing additional structures for this object will likely lead to improvement. For example, Shi et al. (2020) showed that $q(\boldsymbol{u})$ can be parameterised by two sets of inducing points, *orthogonal* to each other, leading to better predictive performance at a much lower compute cost compared to doubling up the inducing points in the standard SVGP approximation. This line of work is complementary to our work here, as it focuses on a different aspect of the posterior, and thus, the two approaches can be combined.

A well-known pathology of variational sparse GP regression is the large estimated observation noise variance (Bauer et al., 2016). It can be partially alleviated by changing the objective function (Jankowiak et al., 2019) or mixing separate schemes for learning and inference (Li et al., 2023). Our work shows that principled structured variational approximations can also partly address this issue.

## 7 Summary

Approximation schemes using inducing points are the method of choice for scaling GP models to large datasets. We show that (i) these methods can be improved by introducing additional structures in the approximate posterior and (ii) these new structures can be applied to various inference strategies, including PEP and variational inference. The resulting methods show comparable or better predictive performance and smaller hyperparameter estimation biases in many standard regression tasks.

There are several potential future directions. First, we have assumed that the size of the data blocks in a dataset is the same and the data partitioning in the experiments was random, but these can be adjusted based on the data characteristics, potentially tightening the variational objective further. Second, the power hyperparameter $\alpha$ in PEP can be made private per block; this will require an understanding of when variational or EP might work best and how to dynamically select $\alpha$. Third, a full discussion for non-Gaussian likelihoods and models beyond GP regression (e.g., deep GPs, GP latent variable models) and how they benefit from structured approximations is a promising exploratory direction.

## Acknowledgments

We would like to thank the anonymous reviewers for the feedback. TDB would like to thank the National Computational Infrastructure (NCI Australia), an NCRIS enabled capability supported by the Australian Government, for the computing resources.

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

# A Full derivation of the block-diagonal variational bound

We start with a general posterior approximation of the form:

$$q(f) = p(f_{\neq \boldsymbol{f}, \boldsymbol{u}} | \boldsymbol{f}, \boldsymbol{u}) q(\boldsymbol{f} | \boldsymbol{u}) q(\boldsymbol{u}) \tag{15}$$

$$q(\boldsymbol{f} | \boldsymbol{u}) = \mathcal{N}(f; \mathbf{K_{fu}} \mathbf{K_{uu}^{-1}} \boldsymbol{u}, \mathbf{C}) \tag{16}$$

where we have not specified the form of the covariance matrix $\mathbf{C}$. The variational lower bound to the log marginal likelihood is

$$\mathcal{F}(q, \theta) = -\mathrm{KL}[q(\boldsymbol{u}) || p(\boldsymbol{u})] - \int q(\boldsymbol{u}) \mathrm{KL}[q(\boldsymbol{f} | \boldsymbol{u}) || p(\boldsymbol{f} | \boldsymbol{u})] + \int q(\boldsymbol{u}) q(\boldsymbol{f} | \boldsymbol{u}) \log p(\boldsymbol{y} | \boldsymbol{f}) \tag{17}$$

Setting the gradient wrt $q(\boldsymbol{u})$ to zeros gives, $q(\boldsymbol{u}) \propto p(\boldsymbol{u}) \exp[\int q(\boldsymbol{f} | \boldsymbol{u}) \log p(\boldsymbol{y} | \boldsymbol{f})]$. In the regression case, $p(\boldsymbol{y} | \boldsymbol{f}) = \mathcal{N}(\boldsymbol{y}; \boldsymbol{f}, \sigma^2 \mathbf{I}_N)$ and thus,

$$\int q(\boldsymbol{f} | \boldsymbol{u}) \log p(\boldsymbol{y} | \boldsymbol{f}) = \int \mathcal{N}(\boldsymbol{f}; \mathbf{K_{fu}} \mathbf{K_{uu}^{-1}} \boldsymbol{u}, \mathbf{C}) \log \mathcal{N}(\boldsymbol{y}; \boldsymbol{f}, \sigma^2 \mathbf{I}_N) \tag{18}$$

$$= \log \mathcal{N}(\boldsymbol{y}; \mathbf{K_{fu}} \mathbf{K_{uu}^{-1}} \boldsymbol{u}, \sigma^2 \mathbf{I}_N) - \frac{1}{2\sigma^2} \mathrm{trace}(\mathbf{C}). \tag{19}$$

The middle term in the bound can be simplified to,

$$\int q(\boldsymbol{u}) \mathrm{KL}[q(\boldsymbol{f} | \boldsymbol{u}) || p(\boldsymbol{f} | \boldsymbol{u})] = \frac{1}{2} \mathrm{trace}(\mathbf{D_{ff}^{-1}} \mathbf{C}) + \frac{1}{2} \log |\mathbf{D_{ff}}| - \frac{1}{2} \log |\mathbf{C}| - \frac{N}{2}. \tag{20}$$

Substituting this and the optimal $q(\boldsymbol{u})$ back to the bound gives,

$$\mathcal{F} = \log \mathcal{N}(\boldsymbol{y}; \mathbf{0}, \mathbf{Q_{ff}} + \sigma^2 \mathbf{I}_N) - \frac{1}{2} \mathrm{trace}[(\mathbf{D_{ff}^{-1}} + \sigma^{-2} \mathbf{I}_N) \mathbf{C}] - \frac{1}{2} \log |\mathbf{D_{ff}}| + \frac{1}{2} \log |\mathbf{C}| + \frac{N}{2}.$$

When $\mathbf{C} = \mathbf{D_{ff}^{1/2}} \mathbf{M} \mathbf{D_{ff}^{1/2}}$, the collapsed bound above becomes,

$$\mathcal{F}(\theta) = \log \mathcal{N}(\boldsymbol{y}; \mathbf{0}, \mathbf{Q_{ff}} + \sigma^2 \mathbf{I}_N) - \frac{1}{2} \sum_b \left[ \frac{1}{\sigma^2} \mathrm{trace}[\mathbf{m}_b \mathbf{D_{f_b f_b}}] + \mathrm{trace}[\mathbf{m}_b] - \log |\mathbf{m}_b| - N_b \right].$$

We can find the gradient of the bound wrt $\mathbf{m}_b$,

$$\frac{\partial}{\partial \mathbf{m}_b} \mathcal{F} = \frac{1}{2} \left[ \sigma^{-2} \mathbf{D_{f_b f_b}} + \mathbf{I}_b - \mathbf{m}_b^{-1} \right] \tag{21}$$

Setting this to zero gives $\mathbf{m}_b = (\mathbf{I}_b + \sigma^{-2} \mathbf{D_{f_b f_b}})^{-1}$, and the resulting $\mathbf{m}$-collapsed bound:

$$\mathcal{F} = \log \mathcal{N}(\boldsymbol{y}; \mathbf{0}, \mathbf{Q_{ff}} + \sigma^2 \mathbf{I}_N) - \frac{1}{2} \sum_b \log |\mathbf{I}_b + \sigma^{-2} \mathbf{D_{f_b f_b}}|. \tag{22}$$

When the block size is 1, the above bounds become the bounds presented in Titsias (2025); Bui et al. (2025).

We now consider a special case when we let all $\mathbf{m}_b$ matrices to be the same, $\mathbf{m}_b = \mathbf{m}$. The gradient wrt $\mathbf{m}$ in this case is,

$$\frac{\partial}{\partial \mathbf{m}} \mathcal{F} = \frac{1}{2} \sum_b \left[ \sigma^{-2} \mathbf{D_{f_b f_b}} + \mathbf{I}_b - \mathbf{m}^{-1} \right]. \tag{23}$$

This leads to the optimal $\mathbf{m}$, $\mathbf{m} = (\mathbf{I}_b + B^{-1} \sigma^{-2} \sum_b \mathbf{D_{f_b f_b}})^{-1}$, and the corresponding $\mathbf{m}$-collapsed bound,

$$\mathcal{F} = \log \mathcal{N}(\boldsymbol{y}; \mathbf{0}, \mathbf{Q_{ff}} + \sigma^2 \mathbf{I}_N) - \frac{B}{2} \log |\mathbf{I}_b + \frac{1}{B\sigma^2} \sum_b \mathbf{D_{f_b f_b}}|. \tag{24}$$

A special case is when the block size is only 1, we arrive at the spherical diagonal approximation $\mathbf{M} = m\mathbf{I}$ (Titsias, 2025; Artemev et al., 2021). Note that, since the log-determinant is a concave function on the cone of positive definite matrices, we can apply Jensen's inequality to show that the bound above (when all $\mathbf{m}$ blocks are the same) is less tight compared to the bound when all blocks are different.

# B  Power-EP posterior and approximate marginal likelihood

## B.1  Power EP steps

Given a data set of $N$ input-output pairs $\{\boldsymbol{x}_n, y_n\}_{n=1}^N$, we use $M$ pseudo-points $\boldsymbol{y}$ at locations $\boldsymbol{z}$ to approximate the exact posterior. Power-EP posits the following approximation to the joint:

$$p(f, \boldsymbol{y}) = p(f_{\neq \boldsymbol{f}, \boldsymbol{u}} | \boldsymbol{f}, \boldsymbol{u}) p(\boldsymbol{f} | \boldsymbol{u}) p(\boldsymbol{u}) \prod_b p(\boldsymbol{y}_b | \boldsymbol{f}_b) \approx p(f_{\neq \boldsymbol{f}, \boldsymbol{u}} | \boldsymbol{f}, \boldsymbol{u}) q(\boldsymbol{f} | \boldsymbol{u}) p(\boldsymbol{u}) \prod_b t_b(\boldsymbol{u}) = q(f)$$

where we have partitioned the data into $B$ disjoint blocks, $b$ indexes blocks of data and $t_b(\boldsymbol{u})$ are the approximate factors. Crucially, we employ a *structured* conditional approximate posterior $q(\boldsymbol{f} | \boldsymbol{u}) = \mathcal{N}(\boldsymbol{f}; \mathbf{K_{fu}} \mathbf{K_{uu}^{-1}} \boldsymbol{u}, \mathbf{D_{ff}^{1/2}} \mathbf{M} \mathbf{D_{ff}^{1/2}})$. The Power-EP procedure with power $\alpha$ iteratively updates the factors $\{t_b\}_{b=1}^B$ as follows:

1. **Deletion step**: Compute the cavity distribution by removing a fraction $\alpha$ of one approximate factor:

$$q^{\backslash i}(f) \propto \frac{q(f)}{t_i^\alpha(\boldsymbol{u})} = p(f_{\neq \boldsymbol{f}, \boldsymbol{u}} | \boldsymbol{f}, \boldsymbol{u}) q(\boldsymbol{f} | \boldsymbol{u}) \frac{q(\boldsymbol{u})}{t_i^\alpha(\boldsymbol{u})} = p(f_{\neq \boldsymbol{f}, \boldsymbol{u}} | \boldsymbol{f}, \boldsymbol{u}) q(\boldsymbol{f} | \boldsymbol{u}) q^{\backslash i}(\boldsymbol{u}), \quad (25)$$

where $q(\boldsymbol{u}) = p(\boldsymbol{u}) \prod_b t_b(\boldsymbol{u})$ and $q^{\backslash i}(\boldsymbol{u}) = q(\boldsymbol{u}) / t_i^\alpha(\boldsymbol{u})$

2. **Projection step**: First, compute the tilted distribution by incorporating a corresponding fraction of the true likelihood factor:

$$\tilde{p}(f) = q^{\backslash i}(f) p^\alpha(\boldsymbol{y}_i | \boldsymbol{f}_i) = p(f_{\neq \boldsymbol{f}, \boldsymbol{u}} | \boldsymbol{f}, \boldsymbol{u}) q(\boldsymbol{f} | \boldsymbol{u}) q^{\backslash i}(\boldsymbol{u}) p^\alpha(\boldsymbol{y}_i | \boldsymbol{f}_i) \quad (26)$$

Second, project the tilted distribution onto the new approximate posterior using KL divergence:

$$q(f) \leftarrow \arg\min_{q(f)} \mathrm{KL}[\tilde{p}(f) || q(f)] \quad (27)$$

Due to the structure of the approximate posterior, this minimisation is achieved when the moments at the pseudo-inputs are matched: $\mathbb{E}_{\tilde{p}(f)}[\phi(\boldsymbol{u})] = \mathbb{E}_{q(f)}[\phi(\boldsymbol{u})]$, where $\phi(\boldsymbol{u}) = \{\boldsymbol{u}, \boldsymbol{u}\boldsymbol{u}^T\}$ are the sufficient statistics (Bui et al., 2017). In practice, this can be done by using the moment-matching shortcut involving the gradients of the log-normalising constant of the tilted distribution.

3. **Update step**: Compute the new fraction by dividing the new approximate posterior by the cavity:

$$t_{i,\text{new}}^\alpha(\boldsymbol{u}) = \frac{q(f)}{q^{\backslash i}(f)} \quad (28)$$

The factor then is updated using $t_i(\boldsymbol{u}) = t_{i,\text{new}}(\boldsymbol{u})$ or with damping, $t_i(\boldsymbol{u}) = t_{i,\text{old}}^{1-\alpha}(\boldsymbol{u}) \cdot t_{i,\text{new}}^\alpha(\boldsymbol{u})$.

## B.2  Optimal factors

The factors are parameterised as follows,

$$t_b(\boldsymbol{u}) = \mathcal{N}(\boldsymbol{u}; z_b, \mathbf{T}_{1,b}, \mathbf{T}_{2,b}) = z_b \exp(\boldsymbol{u}^T \mathbf{T}_{1,b} - \frac{1}{2} \boldsymbol{u}^T \mathbf{T}_{2,b} \boldsymbol{u}) \quad (29)$$

The posterior distribution over $\boldsymbol{u}$ is therefore $q(\boldsymbol{u}) = \mathcal{N}(\boldsymbol{u}; \mathbf{m}, \mathbf{S})$, where

$$\mathbf{S}^{-1} = \mathbf{K_{uu}^{-1}} + \sum_b \mathbf{T}_{2,b} \quad (30)$$

$$\mathbf{S}^{-1} \mathbf{m} = \sum_b \mathbf{T}_{1,b}. \quad (31)$$

Similarly, the cavity distribution over $\boldsymbol{u}$ is $q^{\backslash i}(\boldsymbol{u}) = \mathcal{N}(\boldsymbol{u}; \mathbf{m}^{\backslash i}, \mathbf{S}^{\backslash i})$, where

$$\mathbf{S}^{\backslash i, -1} = \mathbf{K}_{\mathbf{uu}}^{-1} + \sum_{b \neq i} \mathbf{T}_{2,b} + (1 - \alpha)\mathbf{T}_{2,i} = \mathbf{S}^{-1} - \alpha\mathbf{T}_{2,i} \tag{32}$$

$$\mathbf{S}^{\backslash i, -1}\mathbf{m}^{\backslash i} = \sum_{b \neq i} \mathbf{T}_{1,b} + (1 - \alpha)\mathbf{T}_{1,i} = \mathbf{S}^{-1}\mathbf{m} - \alpha\mathbf{T}_{1,i}. \tag{33}$$

The moments of the tilted distribution (and the new posterior) can be computed efficiently using the following shortcuts,

$$\mathbf{m} = \mathbf{m}^{\backslash i} + \mathbf{V}_{\boldsymbol{u}\boldsymbol{f}_i}^{\backslash i} \frac{d \log \tilde{Z}_i}{d\mathbf{m}_{\boldsymbol{f}_i}^{\backslash i}}, \tag{34}$$

$$\mathbf{V} = \mathbf{V}^{\backslash i} + \mathbf{V}_{\boldsymbol{u}\boldsymbol{f}_i}^{\backslash i} \frac{d^2 \log \tilde{Z}_i}{d(\mathbf{m}_{\boldsymbol{f}_i}^{\backslash i})^2} \mathbf{V}_{\boldsymbol{f}_i\boldsymbol{u}}^{\backslash i} \tag{35}$$

where $\tilde{Z}_i = \int q^{\backslash i}(\boldsymbol{f}_i)p^\alpha(\boldsymbol{y}_i|\boldsymbol{f}_i)d\boldsymbol{f}_i$ is the normaliser of the tilted distribution.

At convergence, the optimal form of $\mathbf{T}_{2,b}$ is rank-$N_b$, $\mathbf{T}_{2,b} = \mathbf{w}_b\mathbf{v}_b^{-1}\mathbf{w}_b^T$, where $\mathbf{w}_b = \mathbf{V}_{\boldsymbol{u}\boldsymbol{u}}^{\backslash b, -1}\mathbf{V}_{\boldsymbol{u}\boldsymbol{b}}^{\backslash b} = \mathbf{K}_{\mathbf{uu}}^{-1}\mathbf{K}_{\mathbf{uf}_b}$, $\mathbf{v}_b = -d_2^{-1} - \mathbf{V}_{\boldsymbol{b}\boldsymbol{u}}^{\backslash b}\mathbf{V}_{\boldsymbol{u}\boldsymbol{u}}^{\backslash b, -1}\mathbf{V}_{\boldsymbol{u}\boldsymbol{b}}^{\backslash b}$, and $d_2 = \frac{d^2 \log \tilde{Z}_b}{d(\mathbf{m}_b^{\backslash b})^2}$.

In the regression case, at convergence, $t_b(\boldsymbol{u}) = \mathcal{N}(\mathbf{K}_{\mathbf{f}_b\mathbf{u}}\mathbf{K}_{\mathbf{uu}}^{-1}\boldsymbol{u}; \boldsymbol{y}_b, \alpha[\mathbf{D}_{\mathbf{ff}}^{1/2}\mathbf{M}\mathbf{D}_{\mathbf{ff}}^{1/2}]_{bb} + \sigma^2\mathbf{I}_b)$. We can check this by computing the contribution of an $\alpha$ fraction of the exact likelihood to the posterior $q(\boldsymbol{u})$,

$$\int q(\boldsymbol{f}_b|\boldsymbol{u})p^\alpha(\boldsymbol{y}_b|\boldsymbol{f}_b)d\boldsymbol{f}_b = \int \mathcal{N}(\boldsymbol{f}_b; \mathbf{K}_{\mathbf{f}_b\mathbf{u}}\mathbf{K}_{\mathbf{uu}}^{-1}\boldsymbol{u}, [\mathbf{D}_{\mathbf{ff}}^{1/2}\mathbf{M}\mathbf{D}_{\mathbf{ff}}^{1/2}]_{bb})\mathcal{N}^\alpha(\boldsymbol{y}_b; \boldsymbol{f}_b, \sigma^2\mathbf{I}_b)d\boldsymbol{f}_b \tag{36}$$

$$\propto \mathcal{N}(\boldsymbol{y}_b; \mathbf{K}_{\mathbf{f}_b\mathbf{u}}\mathbf{K}_{\mathbf{uu}}^{-1}\boldsymbol{u}, [\mathbf{D}_{\mathbf{ff}}^{1/2}\mathbf{M}\mathbf{D}_{\mathbf{ff}}^{1/2}]_{bb} + \sigma^2\mathbf{I}_b/\alpha), \tag{37}$$

which is exactly an $\alpha$-fraction of the optimal factor listed above.

## B.3 Power-EP approximate marginal likelihood

After convergence, Power EP provides an approximate log marginal likelihood:

$$\log Z_{\text{PEP}} = \log \int p(f_{\neq \boldsymbol{f}, \boldsymbol{u}}|\boldsymbol{f}, \boldsymbol{u})q(\boldsymbol{f}|\boldsymbol{u})p(\boldsymbol{u})\prod_b t_b(\boldsymbol{u})df \tag{38}$$

$$= \mathcal{G}(q(\boldsymbol{u})) - \mathcal{G}(p(\boldsymbol{u})) + \frac{1}{\alpha}\sum_b \left[\log \tilde{Z}_b + \mathcal{G}(q^{\backslash b}(\boldsymbol{u})) - \mathcal{G}(q(\boldsymbol{u}))\right] + \frac{1}{\alpha}\log \tilde{Z}_q, \tag{39}$$

where

$$\log \tilde{Z}_b = \log \int q(\boldsymbol{f}_b|\boldsymbol{u})q^{\backslash b}(\boldsymbol{u})p^\alpha(\boldsymbol{y}_b|\boldsymbol{f}_b)d\boldsymbol{f}_b d\boldsymbol{u} \tag{40}$$

$$\log \tilde{Z}_q = \log \int q^{1-\alpha}(\boldsymbol{f}_b|\boldsymbol{u})q^{\backslash b}(\boldsymbol{u})p^\alpha(\boldsymbol{f}_b|\boldsymbol{u})d\boldsymbol{f}_b d\boldsymbol{u} \tag{41}$$

$$\mathcal{G}(q(\boldsymbol{u})) = \frac{M}{2}\log(2\pi) + \frac{1}{2}\log|\mathbf{V}| + \frac{1}{2}\mathbf{m}^{\intercal}\mathbf{V}^{-1}\mathbf{m} \tag{42}$$

$$\mathcal{G}(p(\boldsymbol{u})) = \frac{M}{2}\log(2\pi) + \frac{1}{2}\log|\mathbf{K}_{\mathbf{uu}}| \tag{43}$$

$$\mathcal{G}(q^{\backslash b}(\boldsymbol{u})) = \frac{M}{2}\log(2\pi) + \frac{1}{2}\log|\mathbf{V}^{\backslash b}| + \frac{1}{2}\mathbf{m}^{\backslash b, \intercal}\mathbf{V}^{\backslash b, -1}\mathbf{m}^{\backslash b} \tag{44}$$

In the regression case, following closely the steps in (Bui et al., 2017), we can derive the closed-form approximate log marginal likelihood

$$\log Z_{\text{PEP}} = \log \mathcal{N}(\boldsymbol{y}; \mathbf{0}, \mathbf{Q}_{\mathbf{ff}} + \alpha\text{blkdiag}(\{[\mathbf{D}_{\mathbf{ff}}^{1/2}\mathbf{M}\mathbf{D}_{\mathbf{ff}}^{1/2}]_{bb}\}_{b=1}^B) + \sigma^2\mathbf{I}_N)$$

$$+ \sum_b \left[-\frac{1-\alpha}{2\alpha}\log\left|\mathbf{I}_b + \alpha\frac{[\mathbf{D}_{\mathbf{ff}}^{1/2}\mathbf{M}\mathbf{D}_{\mathbf{ff}}^{1/2}]_{bb}}{\sigma^2}\right| - \frac{1}{2\alpha}\log|\mathbf{I}_b + \alpha(\mathbf{m}_b - \mathbf{I}_b)| + \frac{1}{2}\log|\mathbf{m}_b|\right] \tag{45}$$

### B.4 Extension to classification

Instead of working with individual factors, we can use the *stochastic* Power-EP parameterisation (Li et al., 2015) , i.e., assuming contributions from all blocks to the posterior are equal $t_b(\boldsymbol{u}) = t(\boldsymbol{u})$. In addition, instead of running stochastic Power-EP iteration, we can directly work with $q(\boldsymbol{u}) \propto p(\boldsymbol{u})t^B(\boldsymbol{u})$ and optimise the Power-EP energy, also known as the black-box $\alpha$-divergence objective (Hernández-Lobato et al., 2016). We will explore this direction in future work.

## C Additional experimental results

### C.1 Experimental set-up

In addition to the details in the main text, we provide additional information here. For all experiments involving the block-diagonal matrix $M$, we randomly partitioned the training data into $B$ blocks. In the Snelson, kin40k, and Power-EP experiments, we optimised the collapsed bound using the L-BFGS optimiser. In the block-diagonal experiments with medium-scale datasets, we used the Adam optimiser with a learning rate of 0.005. To initialise the inducing point locations, we picked $M$ random training inputs in the Snelson experiment, and employed k-means clustering for all other experiments. For the later datasets, we used the median distance between the data points to initialise the lengthscales and set the initial observation noise variance to 0.1.

### C.2 Snelson dataset

We compared several sparse variational GP variants, including SGPR, T-SGPR, and BT-SGPR, with $M = 10$ to exact GP regression, and the objective and hyperparameters collected during optimisation are included in fig. 6. We note that, by using structured approximations, (i) the variational bound that is provably tighter for fixed hyperparameters indeed is tighter in practice, and (ii) the observation noise variance (the kernel variance) is smaller (larger).

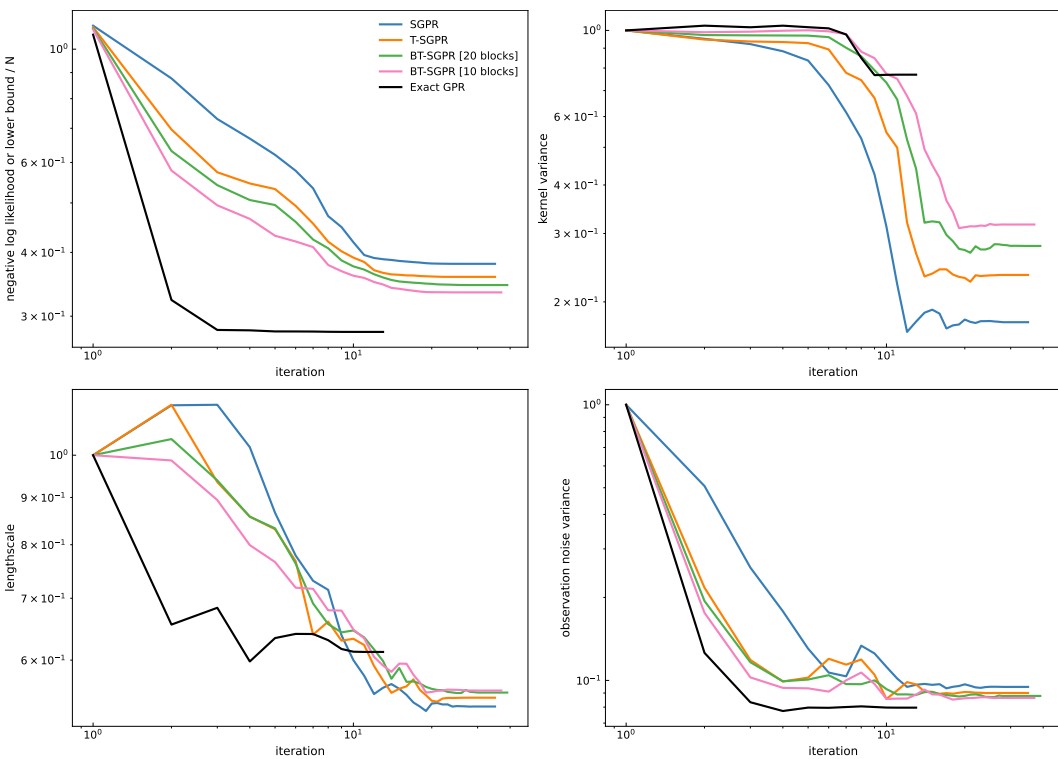

Figure 6: Objectives and hyperparameters provided by sparse variational and exact methods.

### C.3 KIN40K hyperparameters

We include the full results, including the standard errors, for the KIN40K experiment in table 2.

### C.4 Block-diagonal structured variational approximation

In addition to the predictive performance metrics in the main text, we also recorded the estimated hyperparameters when using the new structured variational approximations. These results are included in fig. 7, and agree with observations in smaller datasets (Snelson and kin40k): kernel variance and observation noise variance tend to be larger and smaller, respectively, when using the improved bounds.

### C.5 Power-EP

We include the full results for all five datasets considered in the main text in fig. 8.

Table 2: Exact/approximate marginal likelihoods, predictive performance, and lengthscales given by various methods on 5,000 samples from the KIN40K dataset, including the standard errors across three repeats

| Method | M = 256 | | | | | M = 512 | | | | |
| --- | --- | --- | --- | --- | --- | --- | --- | --- | --- | --- |
| | Obj. | RMSE | LL | $\sigma$ | lengthscales | Obj. | RMSE | LL | $\sigma$ | lengthscales |
| Exact | $-0.656 \pm 0.005$ | $0.117 \pm 0.000$ | $0.796 \pm 0.004$ | $0.001 \pm 0.000$ | | $-0.656 \pm 0.005$ | $0.117 \pm 0.000$ | $0.796 \pm 0.004$ | $0.001 \pm 0.000$ | |
| SGPR | $0.883 \pm 0.006$ | $0.256 \pm 0.001$ | $-0.136 \pm 0.002$ | $0.299 \pm 0.001$ | | $0.660 \pm 0.006$ | $0.215 \pm 0.001$ | $0.022 \pm 0.002$ | $0.252 \pm 0.001$ | |
| T-SGPR | $0.779 \pm 0.006$ | $0.223 \pm 0.001$ | $-0.057 \pm 0.002$ | $0.259 \pm 0.001$ | | $0.515 \pm 0.005$ | $0.184 \pm 0.000$ | $0.115 \pm 0.001$ | $0.206 \pm 0.001$ | |
| BT-SGPR $B = 50$ | $0.752 \pm 0.006$ | $0.217 \pm 0.000$ | $-0.045 \pm 0.002$ | $0.250 \pm 0.001$ | | $0.499 \pm 0.006$ | $0.181 \pm 0.000$ | $0.120 \pm 0.002$ | $0.201 \pm 0.001$ | |
| BT-SGPR $B = 10$ | $0.659 \pm 0.006$ | $0.200 \pm 0.001$ | $-0.032 \pm 0.002$ | $0.227 \pm 0.001$ | | $0.437 \pm 0.006$ | $0.173 \pm 0.000$ | $0.133 \pm 0.002$ | $0.186 \pm 0.001$ | |
| PEP $\alpha = 0.5$ | $0.661 \pm 0.006$ | $0.235 \pm 0.001$ | $-0.015 \pm 0.002$ | $0.225 \pm 0.001$ | | $0.422 \pm 0.005$ | $0.200 \pm 0.001$ | $0.140 \pm 0.002$ | $0.187 \pm 0.000$ | |
| T-PEP $\alpha = 0.5$ | $0.480 \pm 0.005$ | $0.200 \pm 0.000$ | $0.024 \pm 0.002$ | $0.182 \pm 0.001$ | | $0.184 \pm 0.006$ | $0.164 \pm 0.000$ | $0.190 \pm 0.003$ | $0.138 \pm 0.001$ | |

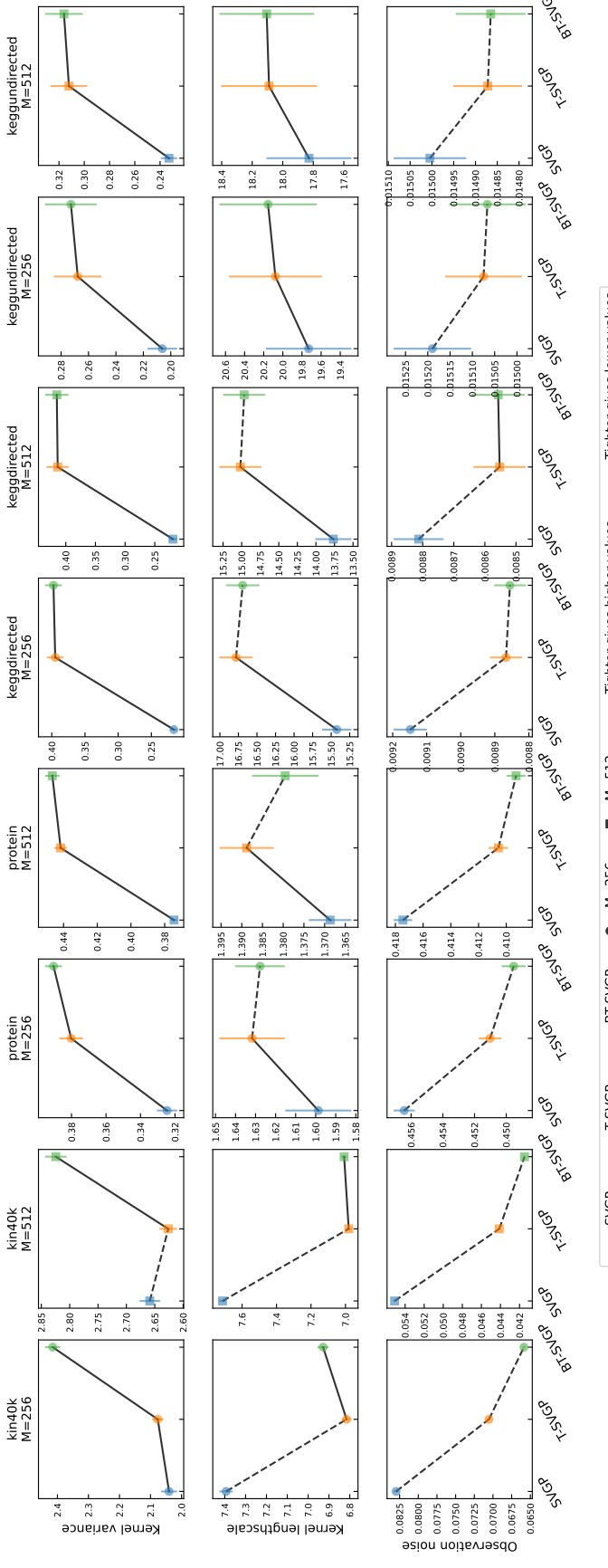

Figure 7: Estimated hyperparameters by using SVGP, T-SVGP and BT-SVGP on four UCI datasets.

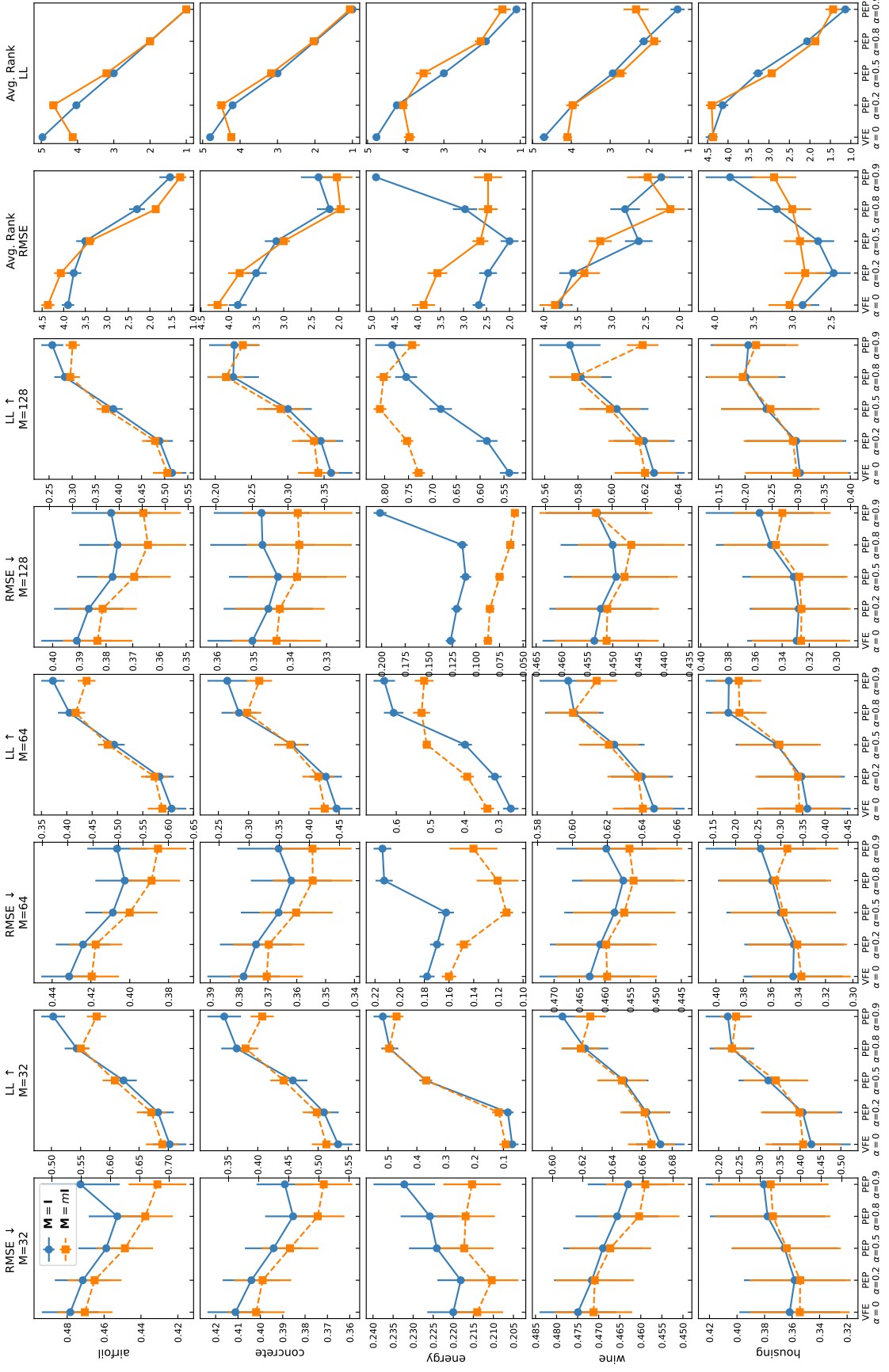

Figure 8: A comparison between $M = I$ and $M = mI$ for Power Expectation Propagation.

