# OpenReview forum: "Sparse Gaussian Processes: Structured Approximations and Power-EP Revisited"
_NeurIPS.cc/2025/Conference — NeurIPS 2025 poster_

### Official Review · Reviewer_XNNT · 2025-06-23

**Clarity:** 2
**Significance:** 3
**Originality:** 3
**Rating:** 4
**Confidence:** 4

**Summary:**

The authors propose a larger variational family for inference in inducing-point-based Gaussian process regression. They show that the optimal posterior in this family leads to a tighter variational lower bound (when using variational inference with a reverse KL-divergence), as compared to several recent works (Titsias, 2025; Bui et al 2025). They then show that this larger variational family can be used within the power expectation propagation framework (Bui et al 2017) to lead to new inference schemes for scalable Gaussian process regression. The authors show the performance, primarily in terms of root mean square error and log likelihood of held-out data.

**Questions:**

- Equations 5 and 6: Why does $\bf{M}$ not appear? Was an optimal form already plugged in given the values of all other parameters? Similarly, in equation 9, have the optimal $\{\mathbf{m}_b\}$ been plugged in?

- In Line 104: Is the square bracket indicating something in the big-O notation? Otherwise simplifying would improve exposition.

- Why would I use equation 5 instead of selecting different values for each block? Would you ever recommend using it? If not, it should be moved to a supplement as essentially an interesting observation.

- `A disadvantage of diagonal and block diagonal structures in M is the expensive predictive covariance.` How expensive would using this predictive covariance be? This should be stated. Could the diagonal elements be calculated efficiently? It looks to me like your experiments only use predictive variances anyway.

- What are the dots above the curves in figure 2? Are these data or the $Z$? This should be stated in the caption.

- What are the solid versus dashed lines in figure 2? This should be stated in the caption or legend.

- What is the (empirical) run time of each method? It seems like (on paper) there is a small amount of additional overhead of the proposed method. But experiments should provide an indication of either run time for each method, or preferably, performance of the methods over time, to give a more detailed sense of the trade-off between performance and run-time for the existing methods versus for the proposed method. This is particularly important because the gains from the proposed method seem to be reasonably small relative to the gains from just increasing the number of inducing points (figure 3).

- Why use the uncollapsed bounds versus the collapsed bounds in 5.2? This requires selection of many more training parameters. See discussion in Section 4.2 of Ober et al 2024.

Ober, Artemev, Wagenländer,  Grobins, van der Wilk. "Recommendations for Baselines and Benchmarking Approximate Gaussian Processes". 2024.

**Ethical Concerns:**

["NO or VERY MINOR ethics concerns only"]

**Final Justification:**

My primary concern in my initial review was that the paper was primarily targeted to an audience very familiar with the sparse Gaussian process literature, and while containing interesting results the paper did not clarify how these could be practically applied to real analyses. In light of the authors proposed addition of a short section clarifying these practical takeaways, as well as proposed changes to improve clarity and ensure notation is defined, I lean towards accepting the paper.

**Limitations:**

Yes

**Paper Formatting Concerns:**

Minor: Line 465. The authors did not delete the instruction block in the checklist that is instructed to be deleted.

**Quality:**

3

**Strengths And Weaknesses:**

## Strengths:

**Connections to existing literature.**
The authors describe how they build on existing work in a clear way. In particular, the authors repeatedly show how their contribution generalizes prior work in this area.

**Some improved performance in regression.**
The authors show some improvements over prior work on several regression datasets.

## Weaknesses:
Overall, my largest criticism of the paper is that it seems largely written for those who are already both experts in, and interested in, inducing-point-based inference in Gaussian process regression. I'm not convinced that the paper is more widely accessible because of notation not being fully introduced and a large number of equations that are not central to the contribution of the paper. I'm also not convinced the authors have fully motivated the contribution for those outside of this community.

**Motivation.**
The motivation for the work is very much presented for someone who already cares a lot about scalable Gaussian process regression. For example the authors propose as a motivating question: ` "Such improvement begs the question: can we achieve even better approximations by considering more expressive structures for M while preserving efficient computation?" ` The paper would be stronger if the authors made and supported a claim about improving a prediction task or data analysis, that is more directly of interest to those who are not already familiar with closely related literature.

**Clarity.**
At the end of section 4, I still don't have a concrete idea of what I am supposed to practically use as a bound. The other have introduced many. Some of these are tractable, some are recovering existing bounds. It should be more clear which bound or bounds the authors think are candidates for obtaining the best posterior/hyperparameter inference.  I would strongly suggest moving some of the bounds (especially intractable bounds) to the supplement, and focusing on bounds you would advocate for the use of in the main text.
Figures: Figure 2 has far too many lines that are not visually distinguishable. I can't get a clear takeaway from looking at it and reading the caption.


**Introducing Notation.**
In many places notation is not clearly introduced.

- Line 27. Notation of u, f, q and p is all introduced at once without sufficient explanation. This is only readable to someone who is already familiar with related work using similar notation.
- Line 51. The authors do not introduce the notation for the covariance function $k_\gamma$.
- Line 57. The authors should state that $\mathbf{u}$ is a random variable when introducing it. Currently it is a bit hard to unpack how $\mathbf{z}$ and $\mathbf{u}$ relate unless someone is already familiar with this notation and related work using it.
- Equation 1. The notation $f_{\neq \mathbf{f}, \mathbf{u}}$ should be introduced. I think it's technically an abuse of notation (two random variables could be equal without being indexed by the same point in input space, and the stochastic process $f$ can't be equal to its evaluation on a finite collection of points since they live on different spaces.) I don't think the notation is unacceptable, but do think it needs an explanation.
- General. The authors introduce lots of lower bounds. It is hard to keep track of them. There is a pattern to the notation using subscript $r, c$. The authors should state what this pattern is explicitly. I assume $r$ is meant to stand for regression (although I wonder if $G$ for Gaussian is more appropriate) and $c$ for collapsed, meaning the optimal posterior is calculated for a fixed set of inducing points and "plugged in".
- Line 114: Differentiate between the optimal $\mathbf{m}$ and a generic $\mathbf{m}$, e.g. use $\mathbf{m}^{\star}$ for the optimal $\mathbf{m}$.
- Equation below line 151: Consider noting the dependence on $\alpha$ in notation on the left hand side.

**Minor:**
Line 77. Remove "Note that". It isn't clear how the author would observe the optimal value of $m_n$ directly from equation 5 and 6.
Line 147. ` The full derivation is rather lengthy and will be included in the appendix`. This should be a cross-reference to a specific section.

**Question 12 in check list is not answered adequately.**
Just because a dataset is publicly available does not mean it isn't licensed. The same applies to code. **The authors should cite datasets and software that are integral to this work.**

---

> ### Author Rebuttal · Authors · 2025-07-27
>
> We thank the reviewer for the detailed feedback and will now address their concerns and questions
>
> *Motivation*: We appreciate the suggestion to improve the motivation. We would like to emphasise that efficient methods for Gaussian processes (GPs) have a notable history of strong presence at NeurIPS and are generally of great interest to machine learners and statisticians, as GPs are widely used in many domains – machine learning, robotics, spatial statistics, and astrophysics, to name a few.
>
> *Clarity*: We kept the key bounds in the main text as we thought it would be useful to describe how the bounds are obtained and related to each other. To improve the clarity, we will summarise them in a subsection in the next version.
>
> *Notation*: We attempt to follow standard definitions and notation in the GP and sparse GP literature (Rasmussen and Williams, 2006; Mathews et al., 2016; Bui et al., 2017), and we are happy to clarify them here as well as in the next version.
>
> * $q$, $p$, $f$, $u$: $q$ and $p$ denote the approximate posterior and prior, respectively. $f$ denotes the entire function, i.e., collection of all function values, $\mathbf{f}$ is the training function values, as explained in line 59, and $\mathbf{u}$ is the inducing outputs, defined in line 57. The difference between $f$ and $\mathbf{f}$ is explained in line 62.
> * $k_\gamma$ is explained in lines 52-53.
> * $\mathbf{u}$: thanks for the suggestion. We will also mention that $\mathbf{u}_m = f(\mathbf{z}_m)$.
> * equation 1: $f_{\neq \mathbf{f}, \mathbf{u}}$ should be read as the function values that are not $\mathbf{f}$ and $\mathbf{u}$.
> * lines 114 and 151: we will fix these.
>
> *Questions*
>
> * Question 1 - equations 5, 6 and 9: Yes, the optimal $\mathbf{M}$ was plugged in.
> * Question 2 - square bracket in line 104: $[N/B]$ simply means the block size.
> * Question 3 - equation 5: Could you please clarify? Did you mean equation 11?
> * Question 4 - expensive predictive covariance: the exact predictive variances are equally expensive, unfortunately, due to the need to invert an NxN matrix, $\mathbf{D}_{\mathbf{ff}}$. Bui et al (2025) showed that dropping the expensive term does not hurt the predictive performance.
> * Question 5 - dots in figure 2: these indicate the inducing inputs $\mathbf{z}$.
> * Question 6 - dotted and solid lines: the solid lines are the predictive mean, and the dashed lines show the 95% credible intervals.
> * Question 7 - runtime: in lines 109-112, page 4, we noted the precise difference between the methods in terms of compute costs. $M$ is typically small, so the overhead is small compared to the overall runtime. For example, on a 2021 Mac laptop, for the kin40k dataset, when M=256, SVGP takes ~15ms for one mini-batch of 500 data points and the overhead when using BT-SVGP is ~1ms,
> * Question 8 - uncollapsed vs collapsed: because of the size of the datasets, the uncollapsed bound is used to enable stochastic optimisation with data minibatching. Using the collapsed bound would be too expensive. Specifically, the sizes of datasets in section 5.2 are 40000 [kin40k], 45730 [protein], 48827 [keggdirected], and 63608 [keggundirected], and thus are out of reach of collapsed bounds.
>
> Thanks again, and please let us know if you have further feedback.

---

> > ### Comment · Reviewer_XNNT · 2025-08-04
> > **Follow up**
> >
> > Thank you to the authors for addressing many of my questions
> >
> > `Question 3 - equation 5: Could you please clarify? Did you mean equation 11?`
> > Yes, I meant equation 11, thank you for catching this.
> >
> >
> > **Practical takeaways.** As a final follow up, I think the paper would be significantly improved by the addition of a more prescriptive paragraph proposing either 1.) a bound that you think is best to use practically or 2.) a workflow (if you do not believe there should be a single "default" bound) to use. As it stands, I think the paper has interesting ideas and results. But I think giving really clear takeaways from the derivations and experiments about which approximation to use, or reasonable, case-dependent heuristics to decide on this, it would improve the practical impact of the paper.
> >
> > If you could outline what a reasonable workflow to select between the bounds you introduce, and when to prefer them to existing bounds, given I have already can specify a prior, dataset and compute budget (and that I can specify any other parameters all sparse GP approaches share), it would help clarify the practical impact for me. If you think this is already in the current draft and I've missed it, a pointer to where it is would be great too.

---

> > > ### Author Response · Authors · 2025-08-05
> > > **Response**
> > >
> > > *Question 3*: Yes, equation 11 is a special case, as we wanted to highlight its connection to the spherical bound in the previous work of Titsias (2025) and Artemev et al. (2021).
> > >
> > > **Practical takeaways**: Thanks for the suggestions. We will add a dedicated "Practical Recommendations" subsection at the end of Section 5 that, based on the experimental results we got, recommends: (i) BT-SVGP [Equation 10] with block size equal to batch size as a default option for variational inference, and (ii) T-PEP with $\alpha = 0.5$ and $\mathbf{M} = m\mathbf{I}$ [Equation 14]. The latter is in agreement with Bui et al. (2017)'s recommendation for intermediate α values and previous empirical observations, e.g., Minka, 2005 [Divergence measures and message passing] and Hernandez-Lobato et al., 2016 [Blackbox alpha divergence minimisation]. We will also link this subsection to the speculations in Section 5, paragraph 1 of Bui et al (2017) on when to prefer a method (VI or EP or in between).
> > >
> > > Thanks again, and please let us know if you have any further suggestions.

---

> > > > ### Comment · Reviewer_XNNT · 2025-08-08
> > > >
> > > > Thank you for the paragraph clarifying practical takeaways and response to my other question. This clarifies my remaining questions.

---

### Official Review · Reviewer_sm4Y · 2025-07-01

**Clarity:** 3
**Significance:** 2
**Originality:** 2
**Rating:** 4
**Confidence:** 4

**Summary:**

The applicability of Gaussian process models is gated by their costly inference, and the go-to solution to GP inference is inducing point sparse variational Gaussian processes (SVGP). SVGP replaces the standard Gaussian process posterior
$$ p(f\vert\mathbf{y}) \propto \prod_{n=1}^N p(y_n \vert f_n) p(f_{\ne \mathbf{f}, \mathbf{u}}\vert \mathbf{f}, \mathbf{u})p(\mathbf{f}\vert \mathbf{u})p(\mathbf{u})$$
with the variational posterior
$$q(f) = p(f_{\ne \mathbf{f}, \mathbf{u}}\vert \mathbf{f}, \mathbf{u}) q(\mathbf{f}\vert \mathbf{u}) q(\mathbf{u}),$$
and minimizes the KL-divergence.
The common wisdom of SVGP replaces $q(\mathbf{f}\vert\mathbf{u})$ with $p(\mathbf{f}\vert\mathbf{u})$, but Bui et al. (2025) and Titsias (2025) suggests that the variational lower bound can be tightened by an additional parameter $\mathbf{M} \in \mathbb{R}^{N\times N}_{\succeq 0}$. Instead of the diagonal scaling matrix $\mathbf{M}$, this paper proposes a block-diagonal structure that tightens the variational lower bound (VLB) without introducing a significant computational overhead, and illustrates theoretically and empirically that the block diagonal structure performs similarly or better to their diagonal matrix alternatives. The paper also draws a connection between SVGP and Power Expectation Propagation, and sees a similar boost in flexibility when a block-diagonal scaling matrix is introduced.

**Questions:**

- I have some trouble understanding the big picture, especially the part about replacing $q(\mathbf{f}\vert \mathbf{u})$ with $p(\mathbf{f}\vert \mathbf{u})$ at prediction. Does the improvement in performance for BT-SGPR mainly comes from better hyperparameter optimization, as the gap between log-likelihood and variational lower bound decreases with the better flexibility of the variational approximation? If we use SGPR with the same hyperparameter value, would the performance be the same?
- I don't understand the statement in line 87-88. How does the flexible covariance matrix $\mathbf{C}$ yield an optimal $q(\mathbf{u})$?
- I understand the variational lower bound formula (e.g., eqs 1-4) are functions related to the variational posterior $q$ and the hyperparameter $\boldsymbol{\theta}$ , but it would be good to state explicitly the arguments to the lower bound functions.
- The block-diagonal structure pre-supposes a partition of data in batches. How would the data be partitioned?

**Ethical Concerns:**

["NO or VERY MINOR ethics concerns only"]

**Final Justification:**

I have interacted with the authors regarding my review, and it has been helpful for me to better understand the paper.

The main concern from my previous assessment is that the paper's contribution seems incremental, especially due to the fact that the random partition of training data introduces a limited performance improvement, and a random partition seems like a suboptimal way to exploit the block-diagonal structure. I am mostly satisfied with the author's promise to look further into this issue and amend the paper accordingly.

My 2nd point of concern about non-conjugate likelihood is not correct, and it does seem that the optimal q(u) with non-conjugate likelihood does not follow the same form, so I think this requires a separate investigation. Therefore, I withdraw my concern in this respect.

**Limitations:**

Yes. I think the authors have adequately addressed the limitation of this paper, as in, the paper's methodological nature has little negative societal impact.

**Paper Formatting Concerns:**

- The checklist included in the paper suggests that the paper provides open access to the code and experimental details, but the submission includes no supplementary code materials and the justification claims that they will be provided "upon acceptance". I think a "yes" answer to this question implies access to the code and experimental settings at submission.

**Quality:**

3

**Strengths And Weaknesses:**

## Strengths
- The paper is well-written and theoretically sound, as the block-diagonal matrix structure generalizes the diagonal scaling matrix, and its added flexibility creates a tighter variational bound. The paper also has a good narrative pace that links a collection of variational lower bounds by emphasizing their connection.
- I am not familiar with the work on power expectation propagation, but the paper's discussion on this topic seems novel and might be of interest to further development in the approximate inference of GPs.
- The paper showcases convincingly that the added flexibility is helpful in obtaining better approximate posteriors and parameter inference via empirical assessments.

## Weaknesses
1. The methodology of introducing an added scaling matrix provides an appealing way to tighten the VLB is an interesting approach, but I am not sure of its applicability: much effort is devoted to learning a flexible parametric form for $q(\mathbf{f}\vert\mathbf{u})$, but it is hard to make use of that as part of the variational predictive distribution (lines 118-123). I understand this drawback applies to all methods that investigates the scaling matrix parametrization, but this paper does not explain further on this point. For example, the second panel of Figure 2 illustrates different approximate posteriors, yet it is hard to see meaningful differences between them.
2. The methodology seems incremental. Replacing the diagonal matrix with a block-diagonal matrix is a natural next step, and it naturally leads to better flexibility. The paper does not discuss the difference in the dimensionality of variational parameters, or its effect on computational complexity. Therefore, if increasing the expressivity only leads to a marginal improvement at the cost of higher computational expense, it would seem that the method has more limited applicability.
3. I think the paper and supplementary materials do not provide adequate details (see the _formatting concerns_ section).

---

> ### Author Rebuttal · Authors · 2025-07-27
>
> We thank the reviewer for constructive feedback. We attempt to answer the reviewer’s questions below.
>
> *Weakness 1*: even when we don’t use $q(\mathbf{f} | \mathbf{u})$ at test time, the test performance can be improved in two ways: more accurate $q(\mathbf{u})$ and better inducing point $\mathbf{z}$, and/or better model hyperparameters. Bui et al (2025) did some ablation studies to isolate the impact, showing that (i) the improved T-SVGP bound improves both $q(\mathbf{u})$ and the hyperparameters, and (ii) using $p(\mathbf{f} | \mathbf{u})$ at test time does not degrade the performance whilst significantly reducing prediction time. We expect the same observation for BT-SVGP and can repeat their studies in the next version. In Figure 2, the key difference between the methods is the hyperparameters, noted in lines 192-194.
>
> *Weakness 2*: We disagree on your comments on applicability. We would like to clarify that the collapsed bound is available (equation 8), and a version that supports data minibatching is also available (equation 10). Thus, using these introduces *no* additional variational parameters compared to SGPR/SVGP and T-SGPR/T-SVGP. The implementation and computational overhead over SVGP and T-SVGP is minimal, as explained in lines 108-112.
>
> *Weakness 3*: Thanks for pointing this out. We will provide code and scripts to reproduce the experiments as promised.
>
> *Question 1*: The performance improvement comes from better $q(\mathbf{u})$, inducing inputs, and hyperparameters. If we use SGPR with the same hyperparameters, the performance of T/BT-SGPR will be different since the $q(\mathbf{u})$ and $\mathbf{z}$ obtained by these methods are different. Please also see our response to Weakness 1.
>
> *Question 2*: in lines 87-88, we assume a general matrix $\mathbf{C}$ and subsequently write down a variational bound that depends on $q(\mathbf{u})$, $\mathbf{z}$ and the hyperparameters. We can optimise this bound wrt $q(\mathbf{u})$ to give the $q(\mathbf{u})$ in line 88, resulting in the collapsed bound on the next line. We hope this is clear.
>
> *Question 3*: Thanks for the suggestion. We will include these arguments.
>
> *Question 4*: As noted in lines 268-270, we used random partitioning for all experiments.
>
> Thanks again, and please let us know if you have further comments.

---

> ### Comment · Reviewer_sm4Y · 2025-08-05
>
> Thank you for your response.
>
> I think my main question relates to my question 1, or more specifically about the author's claim about $ q(\mathbf{u}) $ being more preferable when all other hyperparameters are the same: at prediction, this paper and SVGP/SGPR share the same predictive distribution conditioned on $q$. The collapsed bound (eq. 4) suggests that when the variational posterior is $p(f\vert \mathbf{f}, \mathbf{u})p(\mathbf{f}\vert \mathbf{u}) q(\mathbf{u})$ with a Gaussian likelihood, the optimal $ q(\mathbf{u}) $ is the variational distribution on line 71. If our main objective is to just come up with a variational distribution $q$ and not to optimize eq. 3 with respect to $\theta$, then this goal has already been achieved: the optimal $q$ (line 71) minimizes the KL-divergence. On the other hand, the collapsed bound in e.g., eq. 8 implies the existence of a different $ q(\mathbf{u}) $, but it is only optimal with respect to a different $ q(\mathbf{f}\vert \mathbf{u}) $, which is not used in prediction. Assuming these 2 variational approximations of $ q(\mathbf{u}) $ are different, this would suggest that the author's claim about "a better q" is not accurate, at least when viewed from the perspective of the minimization of KL divergence. Does this make sense?
>
> I understand that the optimization of GP hyperparameters can be improved with a tighter bound, and the experiments do seem to support this claim. My confusion comes from the observation that, after we are done with hyperparameter optimization, could we just use SGPR/SVGP to find the $ q(\mathbf{u}) $, with all other parameters unchanged? Would this result be significantly different?

---

> ### Author Response · Authors · 2025-08-05
> **Response**
>
> We thank the reviewer for engaging in the discussion.
>
> We would like to note that, loosely speaking, there are three *types* of variables in the bound: (1) $q(\mathbf{u})$ [or $q(\mathbf{u})$'s parameters, i.e., mean and covariance, if not collapsed out], (2) the variational parameters $\mathbf{z}$, the locations of the inducing points, and (3) the model hyperparameters such as kernel hypers and noise variance.
>
> One important point the reviewer might have missed is that the optimal form for $q(\mathbf{u})$ depends on $\mathbf{z}$ (in addition to the model hypers), so even if $q(\mathbf{u})$ takes the same analytical form across methods, a tighter bound will give a better $\mathbf{z}$ that gives a different $q(\mathbf{u})$ numerically, and ultimately makes $q(f)$ closer to the true posterior.
>
> We will include an experiment in the next iteration to make this clear.
>
> Please let us know if the above makes sense and if you have any other questions.

---

> > ### Comment · Reviewer_sm4Y · 2025-08-05
> >
> > Thank you for your input.
> >
> > I am aware that a tighter ELBO will potentially lead to a better $\mathbf{z}$ in addition to the hyperparameters $\theta$, which can contribute to better performance overall. However, the premise of my question still stands: once we have made use of the tighter ELBO to find good parameter values for $\mathbf{z}$ and $\theta$, could we plug in these parameters and find a q(u) _within_ the SGPR/SVGP framework, while keeping $\mathbf{z}$ and $\theta$ unchanged? From the perspective of KL minimization, the q(u) obtained in this way would be different and more preferable, since p(f|u) is used at prediction.

---

> > > ### Author Response · Authors · 2025-08-05
> > >
> > > Thanks for the clarification.
> > >
> > > When keeping $\mathbf{z}$ and $\theta$ fixed/unchanged, all variational bounds (SVGP/SGPR or T-SVGP/SGPR or BT-SVGP/SGPR) will end up with the same $q(\mathbf{u})$, as stated on line 71 and line 88. So to answer your earlier point on "existence of a different  $q(\mathbf{u})$": no.
> > >
> > > In addition, our view is that $\mathbf{z}$ and $q(\mathbf{u})$ are variational knobs and thus should be tuned together.
> > >
> > > We hope this is clear, and if not or if you have further questions/requests, please let us know.

---

> > > > ### Comment · Reviewer_sm4Y · 2025-08-06
> > > >
> > > > Thank you for your response, and I am sorry that I overlooked this point previously -- it seems quite obvious now that given different $q(\mathbf{f}\vert \mathbf{u})$, the optimal $q(\mathbf{u})$ stays the same because of how the KL divergence factorizes.
> > > >
> > > > I am, however, in favor of keeping my current assessment because of the following 2 reasons:
> > > >
> > > > - I am somewhat skeptical of the random partitioning of data. Extending from diagonal to block-diagonal matrix is a strict improvement in flexibility, so we should observe consistent improvements upon T-SGPR or when the number of blocks decreases. I think the improvement can be much better illustrated with a more informed partitioning heuristic, as we would intuitively expect that the off-diagonal elements of the covariance matrix are more important when the kernel function implies high correlation. Is it possible to improve the current framework using a simple heuristic that partitions the data into blocks, such that the kernel Gram matrix of each block has high correlation? From my point of view, this can potentially be a simple yet effective heuristic one can only achieve via the block-diagonal structure, yet it remains un-explored.
> > > > - The observation that the optimal q(u) takes the same form across SGPR, T-SGPR and BT-SGPR still stands when the likelihood p(y|f) is non-conjugate, i.e., $ \log q(\mathbf{u}) = C + \log p(\mathbf{u}) + \sum_n \log p(y_n|\mathbf{K}_{f_n \mathbf{u}} \mathbf{K}\_{\mathbf{uu}}^{-1}\mathbf{u}) $. Therefore, it is straightforward enough to extend the current framework to a classification setting by optimizing the un-collapsed bounds. I think the question of non-conjugate likelihood is of interest to scalable GP and is straightforward to implement.
> > > >
> > > > I am thankful for the author's helpful and proactive response, and I do not object strongly to accepting the paper in its current form. However, from my perspective the above 2 points fit well as part of the current work, are straightforward enough to investigate and can go a long way in illustrating to GP practitioners that the block-diagonal structure is impactful.

---

> ### Author Response · Authors · 2025-08-06
>
> We thank the reviewer for the constructive feedback and the good-faith discussion throughout the review process.
>
> On improvement over random partitioning: Your suggestion is very reasonable. We agree that this correlation-based partitioning heuristic is a relatively small adjustment, so we will implement and evaluate this in the next iteration.
>
> On non-Gaussian likelihoods: As noted in Titsias (2025) [Section 3.3], for non-Gaussian likelihoods, moving beyond the *spherical* case ($\mathbf{M} = m\mathbf{I}$) is computationally challenging. We agree that finding a tractable way to combine structured posteriors and non-Gaussian likelihoods is a natural and important next step (see line 273 in the conclusion), but believe it merits a dedicated investigation beyond the scope of the current work.
>
> We hope we have addressed your concerns. Please let us know if you still have any further questions.

---

### Official Review · Reviewer_kA7g · 2025-07-02

**Clarity:** 3
**Significance:** 2
**Originality:** 2
**Rating:** 4
**Confidence:** 2

**Summary:**

This paper focuses on developing a more structured sparse approximation for Gaussian processes by leveraging recent advances in tighter variational bounds (Bui et al., 2025; Titsias, 2025). Specifically, it replaces the diagonal scaling in the posterior covariance with a block-diagonal structure, resulting in a even tighter variational bound. Moreover, this paper extends power expectation propagation (PEP) under this improved bound, or more exactly new posterior approximation. Experiments show that the proposed structured approximation achieves a tighter bound and improved regression performance compared to existing approaches.

**Questions:**

The two methodological foundations for this paper are:
1. **Bui et al., 2025:** tighter variational bound by relax the assumption of $q(f)=p(f_{\neq \boldsymbol{f}, \boldsymbol{u}})p(\boldsymbol{f}|\boldsymbol{u})q(\boldsymbol{u})$ to $q(f)=p(f_{\neq \boldsymbol{f}, \boldsymbol{u}})q(\boldsymbol{f}|\boldsymbol{u})q(\boldsymbol{u})$ where we set $q(\boldsymbol{f}|\boldsymbol{u})=\mathcal{N}(\boldsymbol{f}; \boldsymbol{K_{fu}} \boldsymbol{K_{uu}}^{-1}\boldsymbol{u}, \boldsymbol{D_{ff}}^{1/2}\boldsymbol{M}\boldsymbol{D_{ff}}^{\top/2})$, $\boldsymbol{M}$ is diagonal.

2. **Bui et al., 2017:** PEP updating framework for $q^*(f)=p(f_{\neq \boldsymbol{f},\boldsymbol{u}})p(\boldsymbol{f}|\boldsymbol{u})p(\boldsymbol{}u)\prod_{b}t_b(\boldsymbol{u})$.

If my understanding is correct, the contribution of the current paper lies in extending the above by using a block-diagonal structure for $\boldsymbol{M}$, instead of a diagonal one, and then computing both the corresponding variational lower bound and the PEP objective accordingly.

Hence my concerns are:

1. The proposed use of a block-diagonal structure for the matrix $\boldsymbol{M}$ closely resembles the PITC approximation. In PITC, the conditional distribution is modeled as
$q(\boldsymbol{f}|\boldsymbol{u}) = \prod_b \mathcal{N}(\boldsymbol{f}_b;\boldsymbol{K}\_{\boldsymbol{f}_b}\boldsymbol{K}\_{\boldsymbol{uu}}^{-1} \boldsymbol{u}, \boldsymbol{D}\_{\boldsymbol{f}_b,\boldsymbol{f}_b})$, which leads to a block-diagonal structure in the covariance of $q(f)$. Hence, the current method seems more like an extension of PITC under the tighter bound/posterior approximation, which may weaken the originality of the proposed approach.

2. The current framework appears more like an incremental extension, by replacing the old bound/posterior approximation in PEP with a new one, without offering deeper justification for why PEP is particularly suitable or necessary in this new setting, such as improved robustness, analytical tractability, or computational efficiency.

3. The experimental section lacks some important details. For example, in the 1-D regression task, what kernel was used? How were the training and test sets split? The conclusions drawn from the middle and right plots of Figure 2 are not clearly explained. In Figure 3, what does “tighter is worse” refer to, and under what conditions? The conclusions could be refined to better support the claims.

**Ethical Concerns:**

["NO or VERY MINOR ethics concerns only"]

**Final Justification:**

I had initially questioned whether introducing a block-diagonal structure under the tighter bound framework of Bui et al. (2025) and Titsias (2025) brings a sufficient contribution, given that similar ideas have already appeared in PITC. However, since this work applies the structure at the posterior level, it can be seen as a new direction. However, as I am not fully in the scaling GPs community, I cannot fully assess the impact of this work on the field, and therefore I will lower my confidence.

**Limitations:**

yes, in future work

**Quality:**

2

**Strengths And Weaknesses:**

Although the proposed method is technically complex and requires substantial background knowledge, the overall presentation is smooth and coherent. Still, it may be difficult to follow for readers less familiar with this field.

This paper builds upon recent work on tighter variational bounds (Bui et al., 2025; Titsias, 2025), particularly the inference framework proposed in Bui et al., 2025. The proposed block-diagonal structured approximation closely resembles the Partially Independent Training Conditional (PITC) approximation. While the paper theoretically and  empirically verifies that combining this structured approximation with the tighter bound leads to improved variational bound and performance, the overall contribution appears somewhat incremental, as it primarily extends existing ideas with limited novel theoretical or methodological insights.

---

> ### Author Rebuttal · Authors · 2025-07-27
>
> We thank the reviewer for the questions and feedback. We will now answer the reviewer’s questions.
>
> *Concern 1*: We would like to clarify that our motivation/approach is very different from that of PITC. The PITC approximation is derived from the model/prior modification perspective, whilst our derivation is based on the posterior approximation view (see (Bui et al, 2017) for more discussion). Specifically, PITC modifies the prior so that in the prior, the blocks of function values are conditionally independence given $\mathbf{u}$. Bui et al (2017) showed that this is equivalent to posterior approximation using EP when using multiple function values in a factor. This approach is distinct from what our proposal, we make the approximate posterior richer by utilising $q(\mathbf{f}|\mathbf{u})$ and a structured $\mathbf{M}$ instead of using $p(\mathbf{f}|\mathbf{u})$ (see lines 132-134).
>
> *Concern 2*: We provided supporting evidence in section 5.3 that the new structured posterior approximation improves PEP, in the same way as it does to VFE, in a range of datasets and alphas. We also noted that intermediate alpha values tend to work best. These observations are empirical and, in our view, strong enough to justify the utility of the new approximation. And, generally, we can’t claim that PEP or VFE, or a flexible posterior is guaranteed to lead to a better performance, since the performance of each inference method is dataset and model dependent.
>
> *Concern 3*: Thanks for pointing these out - we will clarify them here and also in the text.
> * kernel: we used the Matern-3/2 kernel.
> * train/test splits: we used the same random train/test spits used in previous benchmark studies such as Yang et al (2015, A la Carte - Learning Fast Kernels) or Wilson et al (2016, deep kernel learning). The link to the splits is provided in the footnote on page 7.
> * middle and right plots in figure 2: The key message we want to get crossed is “hyperparameter optimisation using a more structured approximation tend to result in a smaller noise variance and a larger kernel variance”, as in lines 192-194.
> * figure 3: “tighter” means a method that gives a provably tighter variational bound (for the same hyperparameters), “better” means better lower test RMSE or higher test log-likelihood or lower train lower bound. In this figure, we want to highlight whether moving to using a tighter bound, from SVGP to T-SVGP or from T-SVGP to BT-SVGP, yields a better predictive performance.
>
> Please let us know if you have further questions.

---

> > ### Comment · Reviewer_kA7g · 2025-08-04
> >
> > Thank you for addressing the questions raised above. My main concern remains whether the use of the tighter bound from Bui et al. (2025) and Titsias (2025), combined with the block-diagonal structure similar to PITC, amounts to an incremental improvement. However, since this work applies the block-diagonal approximation in the posterior, unlike PITC which modifies the prior, the contribution may still be meaningful.
> >
> > I suggest adding a discussion paragraph to clarify the similarities and differences with methods like FITC and PITC. Additionally, as the paper assumes considerable background knowledge, it would be helpful to include an additional section in the appendix introducing key concepts such as the tighter bound, DTC, FITC, and PITC to improve accessibility.
> >
> > I am willing to raise my rating to a 4. However, since I am not fully in the scaling GPs community, I will lower my confidence accordingly.

---

> > > ### Author Response · Authors · 2025-08-05
> > > **response**
> > >
> > > We appreciate the suggestion and your willingness to increase the score.
> > >
> > > We will add a dedicated paragraph in Section 6 that explicitly compares and contrasts our method with FITC and PITC. This will highlight how our posterior-based block-diagonal structure differs conceptually and practically from prior-based modifications. We will also include a new appendix section that provides an accessible introduction to background concepts, including the recent tighter bounds and classical sparse GP methods.

---

### Official Review · Reviewer_tnwE · 2025-07-03

**Clarity:** 4
**Significance:** 3
**Originality:** 3
**Rating:** 5
**Confidence:** 3

**Summary:**

Sparse Gaussian processes are popular methods for making GP inference computationally tractable. These methods use a set of small number of function points to approximate GP functions. While they yield tractable inferences, this comes at a cost of expressiveness. The paper proposes an advanced technique to improve the expressiveness of the approximate method while maintaining comparable computational cost. It builds upon an earlier result Bui et al. 2025, a variational approximation method, by further relaxing its diagonal scaling matrix assumption to a block-diagonal matrix. The paper further shows how this approximation technique can also be applied to other approximate methods, such as PEP. Through 1D regression problems, the paper highlights the improvement over standard approximate methods.

**Questions:**

* Eqns 2,3 5,9 show integration but omit the integrand variables.

* Line 255: “however, these methods tend to leverage more dimensions than” any insights why this is the case? Is this related to the expressiveness of the methods? It might be important to see how the block sizes play a role in this as well.

**Ethical Concerns:**

["NO or VERY MINOR ethics concerns only"]

**Final Justification:**

I raised a concern regarding the application of the method beyond 1D datasets. The authors pointed out the experiments they did on >1D datasets. Therefore, I withdraw my concern on this point.


An additional concern I raised was on runtime comparison to see how the computational complexity translates to CPU time.  The authors responded with an example to support the computational advantage of their method.

I also noted the issue on the overall accessibility of the paper to a broader audience, which is also raised by other reviewers. The authors have acknowledged this limitation and noted they will revise the related work section to improve readability.

**Limitations:**

yes

**Quality:**

3

**Strengths And Weaknesses:**

**Strength**

**S1**: The paper deals with improving the accuracy of a popular GP approximation method while still maintaining comparable computation cost. This has a broader utility for both GP community and other ML sub-areas.

**S2**: The paper presents a recipe that generalizes and recovers prior improvements on sparse GP methods (Tistias 2025, Bui 2025, Artemov 2021) from the block-diagonal assumption, giving a clear picture of the proposed method.

**S3**: The experiments show the improvement of using the proposed block diagonal assumption in terms of improved ELBO, and more faithful noise variances (relative to the exact method).


**Weakness**

**W1**: The experiments, focused on 1D regression tasks, make it limited to judging how well this could improve >=2D experiments.


**W2**: While the authors claimed the method “performs similarly or better to diagonal approximation while maintaining comparable computational costs”, there is a lack of experiments to support it. Such experiments could show we can trade accuracy for computational cost (as this work and related works are all striking a good balance between the two).

**W3**: While authors appropriately cited inspiration from Bui et al (2017) framework, I think there could be more discussion on the challenges that come with relaxing the assumption to a block diagonal setting.

---

> ### Author Rebuttal · Authors · 2025-07-27
>
> We thank the reviewer for the constructive feedback. We would like to answer your questions and clarify a few misunderstandings.
>
> *Weakness 1*: Please note that the inputs in our regression tasks (except the Snelson dataset in section 5.1) have >1 dimensions. Specifically, the input dimensions are 8 for kin40k, 9 for protein, 20 for keggdirected, 27 for keggundirect, 5 for airfoil, 8 for concrete, 8 for energy, 11 for wine, and 13 for housing.
>
> *Weakness 2*: in lines 109-112, page 4, we noted the precise difference between the methods in terms of compute costs. M is typically small, so the overhead is small compared to the overall runtime. For example, on a 2021 Mac laptop, for the kin40k dataset, when M=256, SVGP takes ~15ms for one mini-batch of 500 data points, and the overhead for BT-SVGP is 1ms. We will report the full runtimes in the next iteration.
>
> *Weakness 3*: Could you please expand and point us to the specific section in the text?
>
> *Questions*:
> * Equations 2, 3, 5, 9: Thanks! We will fix these.
> * Line 255: Yes, this is related to the tightness of the bound and expressiveness of the variational approximation. We observe that optimising a tighter variational bound yields smaller observation noise, larger kernel variance (please see Figure 7 in the appendix) and more input dimensions being utilised. And thank you for the suggestion on the block-size investigation. We will include this in the next iteration.
>
> Please let us know if you have further questions.

---

> ### Comment · Reviewer_tnwE · 2025-08-04
>
> I thank the reviewers for their clarifying responses.
>
> For weakness 3: to make the paper more complete and more accessible to broader audience, a general note more how the techniques used are similar or different compared to Bui et al (2017), and other related settings may be necessary.
>
> This is inline with `Reviewer XNNT`'s comment on readability,
>
> > Overall, my largest criticism of the paper is that it seems largely written for those who are already both experts in, and interested in, inducing-point-based inference in Gaussian process regression. I'm not convinced that the paper is more widely accessible because of notation not being fully introduced and a large number of equations that are not central to the contribution of the paper. I'm also not convinced the authors have fully motivated the contribution for those outside of this community.

---

> > ### Author Response · Authors · 2025-08-05
> > **Response**
> >
> > We thank the reviewer for the suggestion. As promised to other reviewers, we will add a paragraph in the main text and a section in the appendix to have an accessible introduction to related methods and compare our methods to those in Bui et al (2017). Additionally, we will revise the introduction to better motivate the contribution for the broader ML community.

---

### Author Response · Authors · 2025-08-04
**discussion**

Dear reviewers, we would greatly appreciate it if you could read our rebuttal and let us know if you have any further questions.

---

> ### Author Response · Authors · 2025-08-07
>
> We would like to thank all the reviewers for the questions and discussions so far. If you have any remaining questions/comments/requests before tomorrow, please let us know.

---

### Decision · Program_Chairs · 2025-09-17

**Decision:**

Accept (poster)

**Comment:**

The authors propose a larger variational family for inducing-point-based Gaussian process regression by replacing a diagonal matrix with a block diagonal matrix.  They show that optimization within the family admits tighter variational lower bounds compared to recent works, and describe how to use the power expectation propagation framework for along with this family for a new inference scheme for scalable GPs.  The authors illustrate the performance of the method in terms of RMSE and log likelihood of held-out data.

During the discussion, tmwE had asked for some performance numbers, which were provided to their satisfaction.  Reviewers kA7g and sm4y expressed some skepticism about the contribution of replacing a diagonal with a block diagonal and the connection to the PITC method; kA7g seemed satisfied with the reviewer response, while sm4y remained of the opinion that going from diagonal to block diagonal seemed a bit incremental.  Otherwise, XNNT commented that this paper seems rather inaccessible to those not already steeped in the lore of variational GPs and maybe familiar already with the prior work on PEP; this sentiment was echoed by reviewer TnWE.  The authors responded that they would add a short clarifying section with context, which seemed to satisfy both reviewers.

The main case for accepting this paper is that the change does seem to get good empirical results with a natural generalization of previous work (replacing a diagonal matrix with a block diagonal); I do not necessarily see this as incremental on its own.  The main case against is the presentation, which seems geared primarily toward specialists.